# Accelerated pyro-catalytic hydrogen production enabled by plasmonic local heating of Au on pyroelectric BaTiO$_3$ nanoparticles

Huilin You[1,6], Siqi Li[1,2,3,6], Yulong Fan[2], Xuyun Guo[1], Zezhou Lin[1], Ran Ding[1], Xin Cheng[4], Hao Zhang[5], Tsz Woon Benedict Lo[1,5], Jianhua Hao[1], Ye Zhu[1], Hwa-Yaw Tam[4], Dangyuan Lei[2]✉, Chi-Hang Lam[1] & Haitao Huang[1]✉

The greatest challenge that limits the application of pyro-catalytic materials is the lack of highly frequent thermal cycling due to the enormous heat capacity of ambient environment, resulting in low pyro-catalytic efficiency. Here, we introduce localized plasmonic heat sources to rapidly yet efficiently heat up pyro-catalytic material itself without wasting energy to raise the surrounding temperature, triggering a significantly expedited pyro-catalytic reaction and enabling multiple pyro-catalytic cycling per unit time. In our work, plasmonic metal/pyro-catalyst composite is fabricated by in situ grown gold nanoparticles on three-dimensional structured coral-like BaTiO$_3$ nanoparticles, which achieves a high hydrogen production rate of $133.1 \pm 4.4$ μmol·g$^{-1}$·h$^{-1}$ under pulsed laser irradiation. We also use theoretical analysis to study the effect of plasmonic local heating on pyro-catalysis. The synergy between plasmonic local heating and pyro-catalysis will bring new opportunities in pyro-catalysis for pollutant treatment, clean energy production, and biological applications.

Pyro-catalysis refers to the catalysis triggered by temperature fluctuation induced pyroelectric surface charges in pyroelectric materials[1,2], which is a self-powered catalysis technique by harvesting waste energy from the environment. Recently, pyro-catalysis has attracted increasing attention in clean energy production and reactive oxygen species (ROS) generation[3–9]. Xie et al. reported the coupling between pyroelectric effect and electro-chemical process, where pyroelectric polyvinylidene fluoride and lead zirconate titanate (PZT-5H) polycrystalline ceramic thin films were used as a voltage source to electrolyze water[3]. Xiao et al. presented pyro-catalytic CO$_2$ reduction by using bismuth tungstate nanoplates, where 55.0 μmol g$^{-1}$ methanol production was achieved after 20 thermal cycles between 15 and 70 °C[5]. The pyroelectric induced surface charges are also capable of generating ROS, such as, hydroxyl (•OH), superoxide (•O$_2^-$), singlet oxygen ($^1$O$_2$), and hydrogen peroxide (H$_2$O$_2$)[6–9]. The pyro-catalytic generated ROS can be further used for disinfection and dye treatment. Gutmann et al. reported the impact of thermally excited pyroelectric LiNbO$_3$ and LiTaO$_3$ nano- and microcrystalline powders for the disinfection of *Escherichia Coli* in aqueous solutions, where a high antimicrobial activity was observed[7]. Wu et al. reported the pyro-catalytic decomposition of Rhodamine B (RhB) solution by making use of the hydrothermally synthesized BiFeO$_3$ nanoparticles (NPs)[8]. The RhB dye was almost completely decomposed after undergoing 85 thermal cycles between 27 and 38 °C.

[1]Department of Applied Physics and Research Institute for Smart Energy, The Hong Kong Polytechnic University, Hong Kong SAR, China. [2]Department of Materials Science and Engineering, The Hong Kong Institute of Clean Energy, The City University of Hong Kong, Hong Kong SAR, China. [3]Information Materials and Intelligent Sensing Laboratory of Anhui Province, Key Laboratory of Opto-Electronic Information Acquisition and Manipulation of Ministry of Education, School of Physics and Materials Science, Anhui University, Hefei 230601 Anhui, China. [4]Department of Electrical Engineering, The Hong Kong Polytechnic University, Hong Kong SAR, China. [5]Department of Applied Biology and Chemical Technology, The Hong Kong Polytechnic University, Hong Kong SAR, China. [6]These authors contributed equally: Huilin You, Siqi Li. ✉e-mail: dangylei@cityu.edu.hk; aphhuang@polyu.edu.hk

However, the currently available pyroelectric materials, whose pyro-catalytic capability relies on the variation of ambient temperature, show low pyro-catalytic efficiencies. Under steady state, the short-circuit pyro-current ($I$) available for pyro-catalytic reactions can be calculated as[10]:

$$I = p \cdot A \cdot dT/dt \qquad (1)$$

where $p$ is the pyroelectric coefficient, $A$ is the area of the surface that is normal to the polarization direction, and $dT/dt$ is the temperature change rate. Assuming a pyroelectric coefficient of $10{-}50\,nC\,cm^{-2}\,K^{-1}$ and a typical low environmental temperature ramping rate of $0.1\,K\,s^{-1}$[11-14], the maximum pyroelectric current for pyro-catalytic reaction is only around $1{-}5\,nA\,cm^{-2}$, which is about 2–3 orders of magnitude lower than that of photocatalysts[15]. Simply increasing the temperature ramping rate $dT/dt$ does not help since the total available charge for catalysis during one temperature ramping or cooling is $Q = p \cdot A \cdot \Delta T$, (direct time integral of Eq. 1), showing that the pyro-catalytic reaction depends on the temperature change $\Delta T$ per thermal cycle. Considering that the environmental temperature change is always quite limited, the only way to increase the pyro-catalytic production rate is to increase the number of temperature cycling. However, due to the huge heat capacity of the surrounding media, it is still a great challenge to achieve multiple thermal cycling of the pyro-catalyst within a short time interval using macroscopic heating[16].

To create multiple temperature cycling at the least expense of input thermal energy, it would be ideal to have a localized heat source that only heats up the pyro-catalytic material itself to a certain degree while maintains the surrounding temperature almost unchanged. Plasmonic nanostructures that absorb light and convert it into heat are one of such ideal candidates. The localized heat generated by thermo-plasmonic nanostructures can be easily fine-tuned, turned on or off by external light irradiation, which act as rapid, dynamic, and controllable localized heat sources[17,18]. It has been reported that under the illumination of a 532 nm laser (excitation power of 5 W), there will be a temperature change of around 100 K within a distance of around 1 μm of a Au nanoparticle (around 40 nm) in a timescale of $10{-}100\,\mu s$[19]. Such a large temperature rise in an ultrashort timescale would provide an ideal environment for pyro-catalysis.

Herein, we select barium titanate ($BaTiO_3$) as the model material to investigate the highly efficient and greatly accelerated pyro-catalysis enabled by plasmonic local heating. Among all ferroelectric materials, $BaTiO_3$ exhibits a large pyroelectric coefficient ($p$) of about $20{-}30\,nC\,cm^{-2}\,K^{-1}$ and has been widely investigated as the lead-free perovskite materials for pyroelectric applications[20-22]. As compared with the thick film or bulk counterparts, the use of NPs can greatly enhance the specific surface area of the pyroelectric material and hence increase the available pyroelectric charges[23]. In this work, $BaTiO_3$ NPs were decorated with Au NPs as the plasmonic heat sources, which possess appealing characteristics such as simple structure, easily tunable morphology, and superior photo-thermal conversion efficiency. As one of the most attractive catalytic reactions, hydrogen production from water splitting was used to validate our hypothesis of plasmonic local heating accelerated pyro-catalysis.

In this work, three-dimensional hierarchically structured coral-like $BaTiO_3$ NPs were first synthesized via a hydrothermal method and then coated with in situ grown Au NPs (named as Au/$BaTiO_3$ hereafter). The plasmonic/semiconductor nano reactors have demonstrated an accelerated pyro-catalytic hydrogen production rate of around $133.1 \pm 4.4\,\mu mol\,g^{-1}\,h^{-1}$ by the thermos-plasmonic local heating under irradiation of a nanosecond laser at the wavelength of plasmonic resonance of Au NPs (532 nm). The extremely rapid heating and cooling enabled by the plasmonic local heating and subsequent environment cooling will bring new opportunities for the applications of efficient pyro-catalysis in biological treatment, clean energy production and pollutant removal.

## Results
### Characterization of material

As shown in Fig. 1a, all the diffraction peaks observed in X-ray diffraction (XRD) patterns can be assigned to a pure perovskite phase with a point group 4 mm (JCPDS Card No. 05-0626). The diffraction peaks suggest a tetragonal phase of the as-synthesized $BaTiO_3$ sample, implying that it is ferroelectric[24,25]. The tetragonality ($c/a$ ratio of the lattice parameters) of the $BaTiO_3$ NPs can be estimated from the split of the {200} peaks in the XRD pattern (Fig. 1b), which is around 1.003 (Supplementary Table 1). This value is smaller than the bulk counterpart due to the size effect[26] that usually suppresses the ferroelectricity at the nanoscale range. The ferroelectricity of the synthesized $BaTiO_3$ sample is further verified by piezoresponse force microscope (PFM) in a dual AC resonance tracking mode. As shown in Supplementary Fig. 1, the butterfly-like hysteresis loops and phase switching of 180° demonstrate the ferroelectricity of the $BaTiO_3$. To gain a complete understanding of the band structure, ultraviolet-visible (UV-Vis) diffuse reflectance spectra and ultraviolet photoelectron spectroscopy (UPS) spectra were conducted, as shown in Fig. 1c and d. According to Supplementary Eqs. 3 and 4, the band gap of our synthesized $BaTiO_3$ is 3.23 eV (inset of Fig. 1c), corresponding to a photon energy of ultraviolet light. In Fig. 1d, the low binding energy region ($E_{low\text{-}binding}$) and the secondary electron cut-off energy ($E_{cutoff}$) observed from the UPS spectra of $BaTiO_3$ are 4.38 and 19.02 eV, respectively. Through UPS measurement, the valance band maximum (VBM) can be estimated by, VBM = $h\nu - (E_{cut\text{-}off} - E_{low\text{-}binding})$, where $h\nu$ is photon energy (21.22 eV). Thus, the valence band edge and conduction band edge of $BaTiO_3$ sample are 2.14 and −1.09 eV, respectively, suggesting that the synthesized $BaTiO_3$ material has a suitable energy band structure for water splitting. The scanning electron microscopy (SEM) image (Fig. 1e) shows that the as-prepared $BaTiO_3$ NPs possess a morphology of three-dimensional hierarchically structured coral-like shape with an average size of several hundred nanometers. The high-resolution transmission electron microscopy (HRTEM) image (Fig. 1f) and high-angle annular dark field-scanning TEM (HAADF-STEM) image (Supplementary Fig. 2) of the NPs show three characteristic lattice fringes, which agree well with the $d$-spacings of the (100), (0$\bar{1}$1) and ($\bar{1}$1$\bar{1}$) planes of $BaTiO_3$. Its selected area electron diffraction (SAED) pattern (inset of Fig. 1f) is clear and sharp, manifesting high crystallinity of the synthesized $BaTiO_3$ NPs that agrees well with the XRD pattern (Fig. 1a). The specific surface area of the synthesized $BaTiO_3$ NPs is around $38.85\,m^2/g$, according to Brunauer-Emmett-Teller (BET) surface area measurement (Supplementary Fig. 3).

The SEM and TEM image of Au/$BaTiO_3$ hybrid nanostructures are shown in Fig. 2a and b, respectively. The in situ synthesized Au NPs possess a nanosphere morphology with an average size of around 18 nm (Supplementary Fig. 4). The SAED pattern (inset of Fig. 2b) along the zone axis of [011] also verifies the successful synthesis of Au/$BaTiO_3$ hybrid nanostructures. The HRTEM image (Fig. 2c) of the nanoparticle shows the characteristic lattice fringes that agree well with the $d$-spacings of the (100), (0$\bar{1}$1) and ($\bar{1}$1$\bar{1}$) planes of $BaTiO_3$ and those of (200), ($\bar{1}$11) and ($\bar{1}$1$\bar{1}$) planes of Au. The HAADF-STEM image of the $BaTiO_3$ NPs after the growth of Au NPs is presented in Fig. 2d, where the three characteristic $d$-spacings of the (100), (0$\bar{1}$1) and ($\bar{1}$1$\bar{1}$) planes of $BaTiO_3$ can be clearly seen. This reveals that the growth of Au NPs on the surface of $BaTiO_3$ does not destroy the structure of $BaTiO_3$. The HAADF-STEM image and the associated elemental mappings of Ba, Ti, O and Au (Fig. 2e) show uniform distribution of these elements. The successful synthesis of Au decorated $BaTiO_3$ NPs could also be verified via UV-visible spectra in Supplementary Fig. 5, where the characteristic absorption peak of Au NPs was found to locate at around 532 nm.

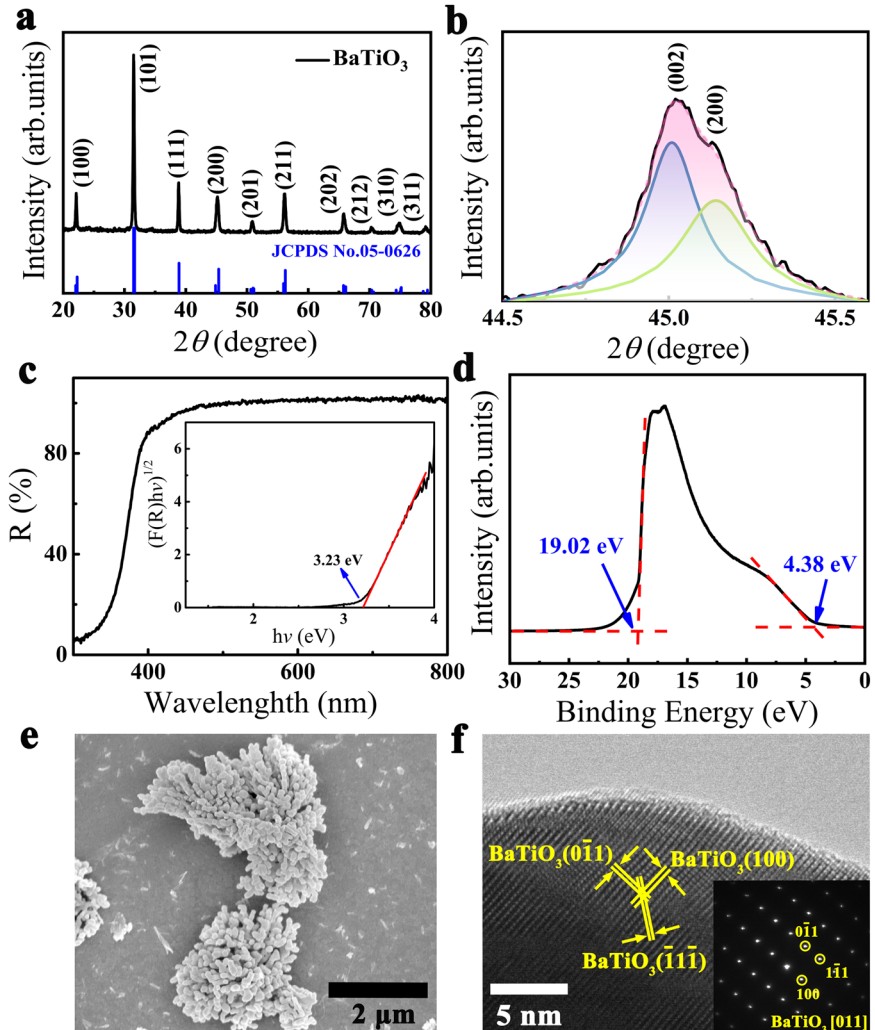

**Fig. 1 | Characterizations of as-prepared BaTiO₃ NPs. a** XRD spectra. **b** Enlarged (002) and (200) XRD peaks. **c** UV-Vis diffuse reflectance spectra with Tauc's plot as inset. **d** UPS. **e** SEM image. **f** HRTEM image. The inset in (**f**) is the SAED pattern along [011] zone axis. Source data are provided as a Source Data file.

## Pyro-catalytic hydrogen evolution from water splitting

Figure 3a reveals the thermo-plasmon induced pyro-catalytic hydrogen generation by Au/BaTiO₃ NPs, where the total hydrogen production under the irradiation power of 786 mW cm⁻² for 60 min is up to 133.1 ± 4.4 μmol g⁻¹. The H₂ and O₂ generated from full water splitting by Au/BaTiO₃ NPs under the irradiation of a 532 nm nanosecond laser is shown in Supplementary Fig. 6 and the results are compared with other reported data (Supplementary Table 2). The morphology of the BaTiO₃ NPs after 90 min pyro-catalysis (Supplementary Figs. 7 and 8) shows almost no change, indicating that the Au/BaTiO₃ NPs have good stability toward pyro-catalytic hydrogen production.

The control experiment in Fig. 3b shows that the BaTiO₃ NPs do not present any noticeable hydrogen production via photocatalysis due to its large band gap of 3.23 eV (Fig. 1c). It can be seen from Fig. 3b that Au/BaTiO₃ NPs exhibit much higher hydrogen production rate than that of Au NPs alone, which indicates that the plasmonic induced catalytic water splitting (such as the reported water dissociation by plasmonic induced hot carriers)[27,28] makes a minor contribution in the present case. From Fig. 3b, it can also be found that, for Au NPs physically mixed with BaTiO₃ NPs, the catalytic water splitting is slightly less than pure Au NPs, indicating that, in this physical mixture of NPs, the catalytic water splitting mainly comes from Au NPs and the light scattering of BaTiO₃ NPs may even slightly degrade the catalytic performance. This result emphasizes the importance of good thermal

contact between Au NPs and BaTiO₃ NPs in order to realize the plasmonic local heating accelerated pyro-catalytic water splitting. It can also be concluded that, for Au/BaTiO₃ NPs, the pyro-catalytic H₂ production makes the most significant contribution to the overall hydrogen production, rather than photocatalysis and hot electron driven water splitting.

Since the total available charge for pyro-catalysis during one temperature ramping or cooling is $Q = p \cdot A \cdot \Delta T$, to have a large hydrogen production, not only the temperature change ($\Delta T$) should be high, but also the number of thermal cycling should be enough. The nanosecond pulsed laser has both high peak power density to generate a sufficiently large $\Delta T$ through plasmonic local heating (which will be discussed below), and enough thermal cyclings per unit time (10 cycles per second in the current work, corresponding to a laser pulse repetition rate of 10 Hz). With increasing average power density of the laser pulse, $\Delta T$ will be increased through the plasmonic photo-thermal conversion[29], leading to higher hydrogen production (Fig. 3c).

To reveal the importance of thermal cycling, we compare the hydrogen production by illuminating the Au/BaTiO₃ NPs with different light sources. Though the average power density of a continuous wave (CW) laser (14.15 W cm⁻²) and a xenon lamp (694 mW cm⁻²) are much higher or very close to that of a nanosecond laser (786 mW cm⁻²), no detectable hydrogen production under the illumination of these two light sources can be found (Fig. 3d). This can be ascribed to the fact

that the temperature change of the plasmonic NPs induced by CW laser or xenon lamp irradiation is too low. For Au NPs with a radius of 10 nm to have 5 °C increase in temperature, a power density of $1 \times 10^5$ W cm$^{-2}$ is needed[30]. Besides, the raised temperature will be kept almost constant under the continuous illumination of a CW laser or a xenon lamp. This is in strong contrast to the case of a pulsed laser. For a nanosecond laser, due to the fast energy releases in several nanoseconds, a very high peak power density up to $5.09 \times 10^6$ W cm$^{-2}$ (average power density of 786 mW cm$^{-2}$) can be achieved within 24 ns, resulting in a large temperature rise on the Au/BaTiO$_3$ NPs within a short time. Moreover, the time interval between successive laser pulses of the nanosecond laser is much longer than that of the laser pulse itself, which allows enough time for the heated BaTiO$_3$ NPs to cool down by the surrounding liquid, and make them ready for the next thermal cycle stimulated by the next laser pulse. It can be predicted that, by increasing the laser pulse repetition rate, the hydrogen production rate can be further increased due to increased number of thermal cycling per unit time.

### Thermal simulation

To better understand the mechanism of plasmon induced localized pyro-catalysis, commercial full-wave finite element method simulations by COMSOL RF and Multiphysics 5.5 modules were performed. The simplified structural model is depicted in Supplementary Fig. 9, where the temperature distribution profile of the Au nanosphere (9 nm in radius) grown on BaTiO$_3$ NP can be obtained.

The heat absorbed by BaTiO$_3$ NP or surrounding water largely comes from calefactive Au NP through heat conduction. Note that although thermal convection does exist in any stirred system, the globally unchanged temperature (will be demonstrated in the following part) of the surrounding water allows us to neglect this heat transfer process to simplify our quantitative study on the effect of local heating induced pyroelectricity. A structural model for the calculation of temperature distribution of Au decorated BaTiO$_3$ NP is shown in Fig. 4a, where an average value of 9 nm is typically taken for the radius of Au NP, according to the experimental value (Supplementary Fig. 4). The point (0, 0, 0) is defined as the surface position of BaTiO$_3$ just beneath the center of Au nanosphere. P-BaTiO$_3$ denotes the region of BaTiO$_3$ NP (the cylindrical region with a length of 200 nm and a radius of 50 nm), and W-BaTiO$_3$ is the whole BaTiO$_3$ NP (a cylinder with a length of 1 μm and a radius of 50 nm). Surrounding water denotes the water sphere which surrounds the whole Au/BaTiO$_3$ with diameter of 20 μm. Figure 4b and c show the time evolution of the temperatures at different regions of a Au/BaTiO$_3$ NP and the surrounding water. The effect of the radius of Au nanosphere on the temperature variation can be seen in Supplementary Fig. 10, which agrees with other groups' observations[30]. It can be concluded from Fig. 4 and Supplementary Fig. 10 that the temperature changes of Au nanosphere and P-BaTiO$_3$ increase with increasing radius of decorated Au nanospheres, while those of W-BaTiO$_3$ and surrounding water show no obvious increase. This result is verified by direct monitoring of the water temperature during 90 min of nanosecond laser irradiation (Supplementary Fig. 11) with a thermometer. On the BaTiO$_3$ NP, the temperature rise at a position closer to the Au nanosphere is also larger (Supplementary Fig. 10e).

Moreover, when the lase pulse is off, the Au and BaTiO$_3$ NPs cool down to room temperature in a period of about 50 ns (Fig. 4b) due to the small total heat capacity of the localized plasmon heated region as compared with that of the vast surroundings. It should be emphasized that no matter heating or cooling, pyro-catalytic reaction will take place so long as the temperature is changed. Therefore, the plasmonic localized heating enables multiple thermal cycles within a short time period due to its rapid heating and cooling processes, which is advantageous to the efficient increase of the overall pyro-catalytic reaction product and reaction rate.

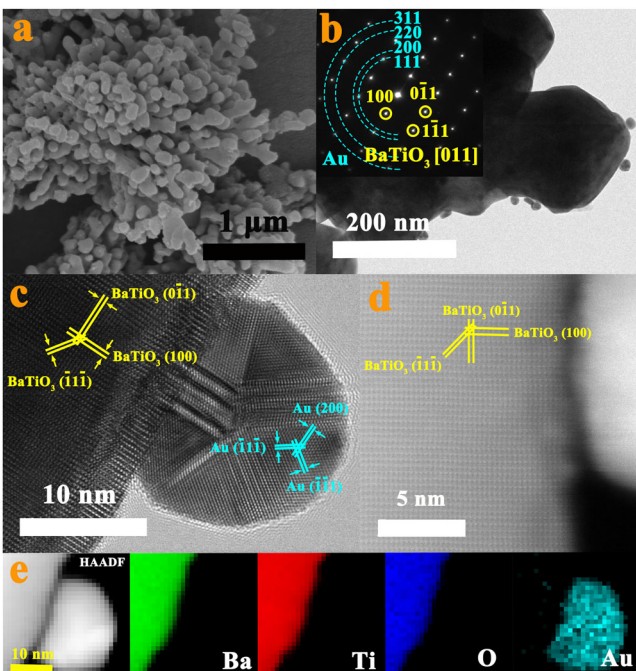

**Fig. 2 | Morphology characterizations of Au/BaTiO$_3$ NPs. a** SEM image. **b** TEM image with SAED pattern as inset. **c** HRTEM image. **d** HAADF-STEM image with atomic structure as inset. **e** HAADF-STEM image and corresponding elemental mapping of Ba, Ti, O, and Au elements.

## Discussion

A schematic illustration of pyro-catalytic hydrogen generation of Au/BaTiO$_3$ NP driven by surface plasmon induced local heating is shown in Fig. 5. Under illumination, the surface plasmon resonance of a Au NP induces a rapid increase in the local temperature of its attached BaTiO$_3$ NP. The uncompensated pyroelectric charges on the BaTiO$_3$ surface can react with surrounding water molecules to generate hydrogen and oxygen, which is discussed in detail in Supporting Information (Supplementary Fig. 12). When the illumination is off, the BaTiO$_3$ will undergo a cooling cycle with rapid dissipation of the generated heat into the surrounding water. Again, the uncompensated pyroelectric surface charges during the cooling cycle will participate in the pyro-catalytic water splitting. Moreover, the internal electric field built up from the surface pyroelectric charges can further facilitate the charge separation and charge transfer for pyro-catalytic hydrogen and oxygen production.

In addition to the pyro-catalysis of BaTiO$_3$ and hot electron catalysis of Au, thermoelectric effect could also contribute to the overall catalytic reaction of water splitting. Thus, we perform thermal calculation by COMSOL Multiphysics 5.5 to estimate the thermoelectric effect of Au/BaTiO$_3$. A simplified structural model (Supplementary Fig. 13) is used to calculate the quantity of charges released due to the thermoelectric effect (Supplementary Fig. 14).

As a comparison, for pyroelectric materials, the quantity of electrons induced by pyroelectric effect can be expressed by as[31],

$$Q = p \cdot \int \triangle T \left( \hat{\mathbf{n}} \cdot \hat{\mathbf{P}} \right) dA \qquad (2)$$

where $p$ is the pyroelectric coefficient with a value between 20 and 30 nC cm$^{-2}$ K$^{-1}$, $\triangle T$ is the temperature rise in heating (or drop during cooling), $\hat{\mathbf{n}}$ is the unit vector along the surface normal direction, and $\hat{\mathbf{P}}$ is the unit vector along spontaneous polarization direction. The surface integral of $\triangle T(\hat{\mathbf{n}} \cdot \hat{\mathbf{P}})dA$ with a corresponding spontaneous polarization perpendicular to the axial direction is

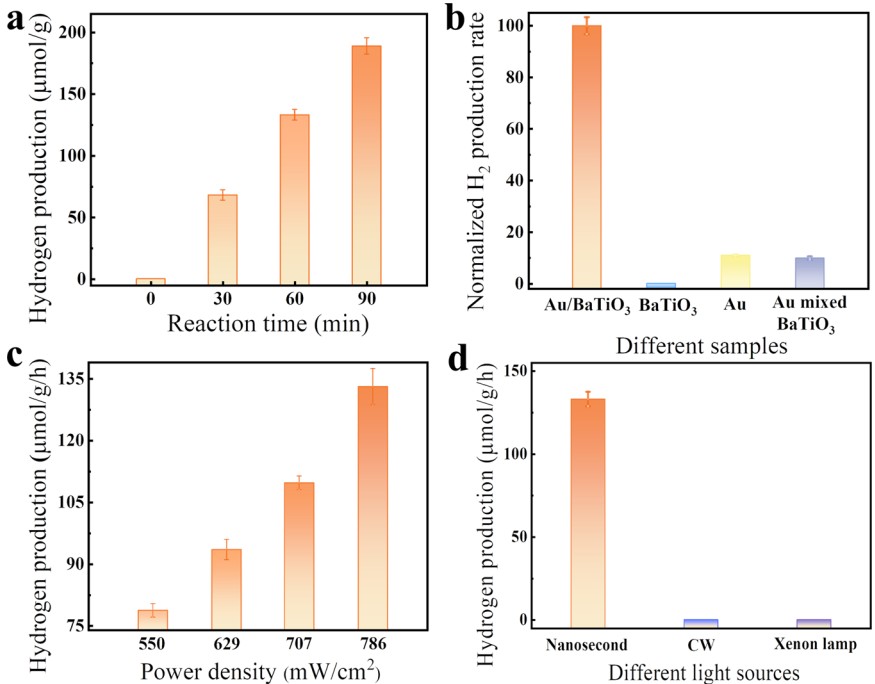

**Fig. 3 | H₂ generation from water splitting by Au/BaTiO₃ NPs. a** Hydrogen generation under different reaction time. **b** Normalized H₂ production rate (normalized to the production rate of Au/BaTiO₃ NPs) of different samples under the irradiation by a 532 nm nanosecond laser. **c** Hydrogen generation under the irradiation of a 532 nm nanosecond pulsed laser with different power densities. **d** H₂ generation by Au/BaTiO₃ NPs illuminated under different light sources. The error bars represent the standard deviation of three parallel experiments. Source data are provided as a Source Data file.

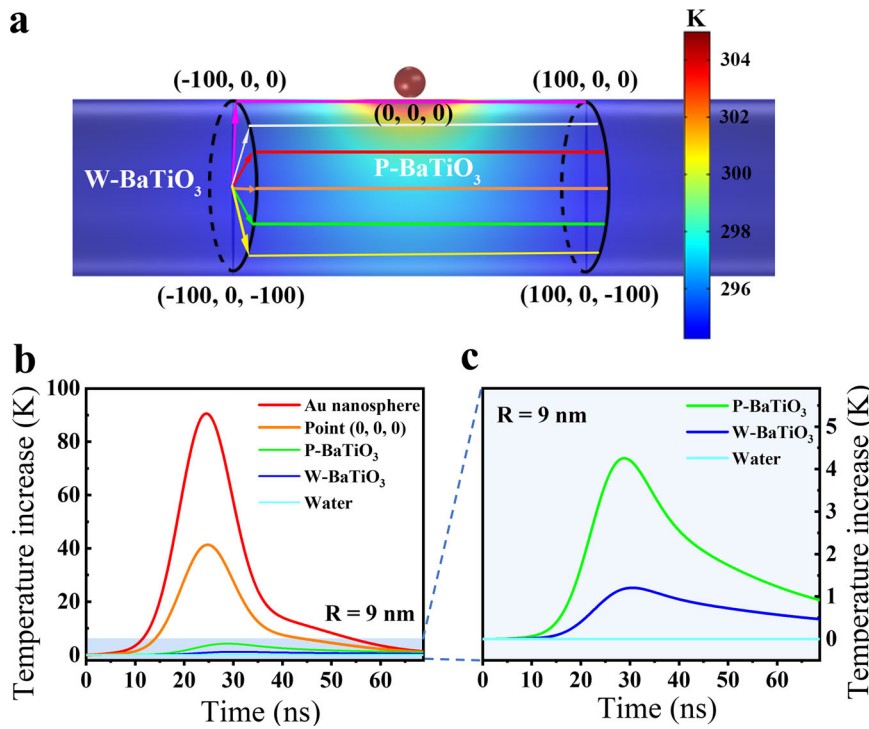

**Fig. 4 | Thermal simulation. a** Temperature distribution of the structural model of a Au NP (9 nm in radius) suspending over a BaTiO₃ cylinder with length of 1 μm and radius of 50 nm at the moment with highest temperature inside Au NP. **b** Time evolution of the temperature of point (0, 0, 0) and the average temperature of Au NP, region of P-BaTiO₃, region of W-BaTiO₃, and surrounding water. **c** Enlarged time evolution of the average temperatures of the regions of P-BaTiO₃, W-BaTiO₃, and surrounding water. Source data are provided as a Source Data file.

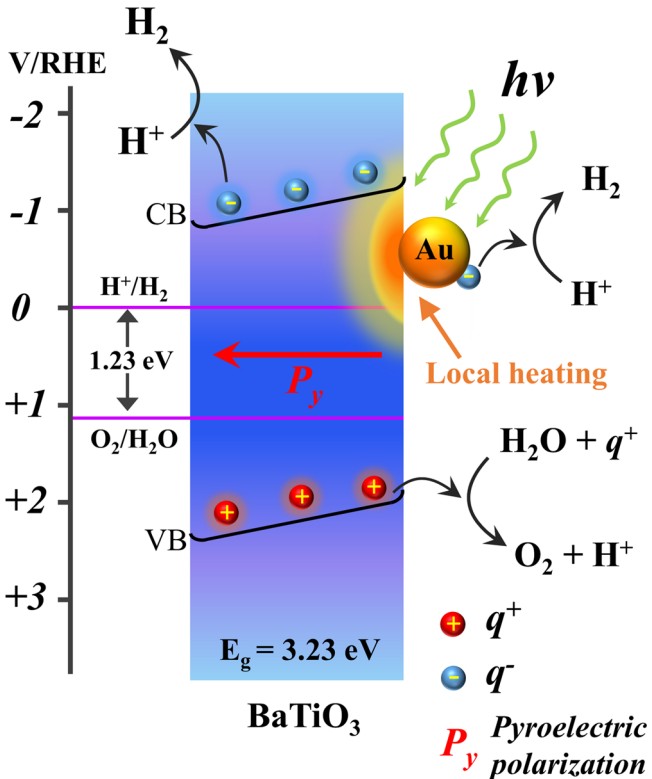

**Fig. 5 | Pyro-catalytic mechanism.** Schematic illustration of pyro-catalytic hydrogen generation of Au/BaTiO$_3$ NP driven by surface plasmon local heating.

shown in Supplementary Fig. 15. Taking the pyroelectric coefficient ($p$) of 30 nC cm$^{-2}$ K$^{-1}$ as an example, we can easily calculate that the pyroelectric charges in one heating/cooling cycle are 5.12 × 10$^{-17}$ C, which are much larger than those due to thermoelectric effect (Supplementary Fig. 14). Therefore, the thermoelectric effect due to the non-uniform local heating by the plasmonic NPs can be neglected. Here, the photocatalytic contribution to the water splitting can also be excluded since BaTiO$_3$ has a large band gap located in the ultraviolet range.

To evaluate the pyro-catalytic hydrogen generation performance of BaTiO$_3$, we calculate the the total available pyroelectric charges using Eq. 2 (see supporting information for detailed calculation). In general, to increase the pyro-catalytic H$_2$ generation rate, we can either reduce the size of BaTiO$_3$ NP or replace the current laser source by one with a higher repetition rate. Our above discussion is based on the assumption that there is only one Au NP on a single BaTiO$_3$ particle. If there are multiple optically isolated Au NPs decorated on a single BaTiO$_3$ particle, the heating effect will be algebraically augmented and the pyro-catalytic H$_2$ production rate will be further increased. If some of the Au NPs are closely packed to trigger electromagnetic coupling, then the heating effect will be more than simple algebraic summation. The above estimation does not take into consideration the possible factors that may restrict the maximum achievable pyro-catalytic H$_2$ production rate, such as surface charge loss by capacitive effects, insufficient absorption of the incident light, and charge recombination, etc[32,33]. We also want to emphasize that the pyro-catalytic hydrogen production reported here cannot be directly benchmarked to the state-of-the-art photocatalytic hydrogen production from water splitting since the underlying mechanisms are different.

Apart from hydrogen production, hydroxyl radicals are also detected during the pyro-catalysis (Supplementary Fig. 16), which can be used for biological applications such as cancer therapy[34–37].

Localized pyro-catalysis is effective only in nanoscale range and does not affect the surroundings, which is an attractive feature for achieving accurate treatment in tumor cell, leading to minor side effects. The wavelength of the excitation light can also be adjusted to near infrared band (biological window) through changing the morphology of Au NPs, for instance, utilizing gold nanorods. Decreasing the diameter of laser beam to μm can largely decease the power needed for plasmon induced pyro-catalysis.

In summary, we have demonstrated the greatly accelerated pyro-catalytic hydrogen production by coupling pyroelectric material with thermal-plasmonic one: three-dimensional hierarchically structured coral-like BaTiO$_3$ NPs were capped by in situ grown Au NPs. A high hydrogen production rate of 133.1 ± 4.4 μmol g$^{-1}$h$^{-1}$ was achieved under the irradiation of a 532 nm nanosecond laser with 0.5 W optical power. The pulsed laser irradiation brings about a dramatically rapid local heating under pulsed excitation, and a fast cooling during pulse off period, thus greatly promoting and accelerating the overall pyro-catalytic hydrogen production. The synergy between plasmonic local heating effect and pyro-catalysis will open up new avenues for efficient catalysis for biological applications, clean energy production and pollutant treatment, etc.

## Methods

### Synthesis of BaTiO$_3$ and Au decorated on BaTiO$_3$ nano-particles (NPs)

BaTiO$_3$ NPs were synthesized through a hydrothermal method[38]. Typically, 1.25 g titanium dioxide (TiO$_2$) (Sinopharm Chemical Reagent Co. Ltd.) and 24.08 g sodium hydroxide (NaOH) (International Laboratory USA) were dissolved into 60 mL of deionized water. After stirring for 30 min, the mixed solution was transferred into a 100 mL Teflon-lined stainless-steel autoclave, heated at 180 °C for 24 h, and then cooled down to room temperature naturally to get sodium titanate (Na$_2$Ti$_3$O$_7$). The obtained material was then dispersed into HCl solution (pH = 1) for 4 h to obtain hydrogen titanate (H$_2$Ti$_3$O$_7$). After repeated wash with deionized (DI) water, H$_2$Ti$_3$O$_7$ was washed to pH = 7 and dried at 60 °C for 12 h. After that, 0.1288 g H$_2$Ti$_3$O$_7$ and 0.95 g barium hydroxide (Ba(OH)$_2$·8H$_2$O) (Sinopharm Chemical Reagent Co. Ltd.) were dissolved in 60 mL of deionized water. The mixed solution was set in a 100-mL Teflon-lined stainless-steel autoclave at 210 °C for 85 min and cooled down to room temperature naturally. Then, barium titanate (BaTiO$_3$) was collected by centrifugation and washed with DI water and ethanol several times until pH = 7. Finally, BaTiO$_3$ catalyst was obtained after dying at 60 °C for 12 h. The Au/BaTiO$_3$ NPs were synthesized in situ via a citrate reduction method. In the synthesis of the Au/BaTiO$_3$ NPs, 16.5 mg BaTiO$_3$ NPs, 750 μL sodium citrate (40 mmol L$^{-1}$) (International Laboratory USA) and 500 μL HAuCl$_4$ (10 mmol L$^{-1}$) (Sigma-Aldrich) were added into 19 mL water. After 30 s ultrasonication, the solution was heated to reach boiling point for several minutes (around 10–12 min) under vigorous stirring to obtain the Au/BaTiO$_3$ NPs (until the color the solution turns to be purplish red). For hydrogen production experiment, 3 mL Au/BaTiO$_3$ NPs solution were collected by centrifugation at 6000 *rpm* and dispersed in 12 mL water. After 10 s ultrasonication, the obtained catalyst was added into the reactor for H$_2$ production. All reagents used as starting materials were of analytical grades.

### Characterization of catalyst

The XRD patterns of synthesized BaTiO$_3$ were recorded by Rigaku SmartLab 9KW X-ray powder diffractometer (scan rate of 0.1°/min, scan range of 20–80 degree and the wavelength of the XRD radiation of $\lambda$-1.54 Å). The morphology of BaTiO$_3$ was studied by a TESCAN MAIA3 SEM, where the catalyst was dispersed on carbon conducting paste. Transmission electron microscopy (TEM) and scanning TEM (STEM) were performed using JEOL JEM-2100F TEM/STEM operated at 200 kV, where the catalyst was dispersed in ethanol and then transfer

into cupper grid. Electron energy-loss spectroscopy (EELS, by Gatan Enfina) mapping was carried out under 200 kV accelerating voltage with a 13 mrad convergence angle for the optimal probe condition. Energy dispersion of 0.5 eV per channel and 21 mrad collection angle were set up for EELS. High-angle annular dark field (HAADF) images were acquired with an 89 mrad inner angle simultaneously. For High-resolution scanning transmission electron microscopy (HRSTEM), the HAADF detector collection inner angle was set to 41 mrad to increase the S/N ratio. Piezoresponse force microscopy (PFM, Asylum MFP 3D Infinity) was used to characterize the ferroelectricity of the $BaTiO_3$ catalyst, where the catalyst was dispersed on Pt coated silicon sheet. UV-Vis Diffuse reflectance spectra was tested by Shimadzu UV-2550 UV-vis spectrophotometer. The UPS was measured by thermo Fisher Nexsa X-ray photoemission spectroscopy, where the catalyst was dispersed on Pt coated silicon sheet (photon energy of 21.22 eV and bias voltage of −10 V). Brunauer-Emmett-Teller (BET) specific surface area analysis was conducted via surface area and porosity analyzer (Micromeritics, ASAP 2020). Power density of one laser pulse was measured via a digital storage oscilloscope (Keysight, Infiniium S-Series, 1 GHz, 20 GSa·s$^{-1}$, 10-bit ADC) with biased silicon detector (EOT, ET-2030, Bandwidth >1.2 GHz, Risetime<300 ps). Hydroxyl was detected via photoluminescence (PL) spectra via Edinburgh FLS920 spectrofluorometer with 321 nm UV light.

## Hydrogen production experiments

The pyro-catalytic hydrogen production of the Au decorated $BaTiO_3$ NPs was evaluated offline. In pyro-catalytic experiment, around 2.62 mg Au decorated $BaTiO_3$ NPs were dispersed in 12 mL deionized water. The aqueous suspension was put into a 30 mL pear-shaped quartz reactor and sealed using septa in advance, which was then evacuated and purged by $N_2$ for about 20 min to completely remove air. A 500 mW 532 nm nanosecond pulsed laser (Continuum, Inlite II) with repetition rate of 10 HZ and pulse width of around 12.7 ns (see Supplementary Fig. 9c for detailed information), a 1 W 532 nm continuous laser (Honkoktech, PSU-H-LED) and a 300 W high-pressure Xenon lamp (Perfect Light, PLS-SXE300) were utilized as light sources. The distance between irradiation source and reactor is 10 cm. To detect the amount of hydrogen production, 300 μL gas component within the reactor was intermittently extracted and injected into a gas chromatograph (Agilent 7890B) with a thermal conductivity detector (chromatographic column: Agilent, length of 30 m and diameter 0.32 mm). 300 μL $N_2$ gas is refilled into reactor after sampling, which was taken into consideration in the calculation. The amount of hydrogen gas produced was calculated using a calibration curve ($y = 137.78x + 18.95$, $R^2 = 0.99962$) of hydrogen concentration versus peak area (working range: 300–15,000 ppm; limit of detection: 50 ppm; limit of quantification: 150 ppm). All the hydrogen production experiments were performed three times per experimental parameter set.

## Simulation details

The optical modelling of the plasmonic local heating effect was carried out by two steps of simulation by using the commercial full-wave Finite Element Method software (COMSOL RF and Multiphysics 5.5). A circularly polarized plane wave was used as the background field to interact with the faceted gold nanospheres with different radii, i.e., 6, 9, and 12 nm suspended over a $BaTiO_3$ cylinder. $BaTiO_3$ nanowire is placed 1 nm below the faceted gold nanospheres. The absorption cross section was thus obtained as the absorbed energy normalized to the incident optical intensity. PML condition was applied to the outer surface of water domain. The plasmonic local heating effect was studied by using the COMSOL Multiphysics 5.5, Heat Transfer module. The geometries and the relative position of the faceted gold nanospheres and $BaTiO_3$ cylinder are all set the same as the optical modelling except that they are surrounded by a water

sphere with 20 μm in diameter, which is large enough to render the average temperature rise negligible during the one-pulse excitation, complying with the experimental observation. The gold nanospheres are modeled as heat sources whose heating power equals to the time dependent laser power density times their absorption cross sections obtained from the optical modelling. The heat transfer coefficient 1000 W m$^{-2}$ K$^{-1}$ of the convective heat flux was assigned to the outer surface of the water sphere, a commonly used value describing the stirring process. The mesh size in the nanosphere and that in the Part $BaTiO_3$ (P-$BaTiO_3$, the cylindrical region with a length of 200 nm and a radius of 50 nm) are set to be a constant (2 nm). The mesh size in the whole $BaTiO_3$ (W- $BaTiO_3$, a cylinder with a length of 1 μm and a radius of 50 nm) and that in the water sphere gradually increase from 2 nm to 20 nm and from 2 nm to 500 nm respectively, to accelerate the simulation. In this study, a point probe at (0, 0, 0) has been used to measure the temperature rise of the point on the $BaTiO_3$ surface but right underneath the gold nanosphere, and four domain probes has been used to monitor the average temperature rises in the gold nanosphere, P-$BaTiO_3$, W-$BaTiO_3$ and the water sphere, respectively. The domain probe can integrate the temperature rise within the probed domain and divide it by the volume, thus giving us directly the average temperature rises during the whole heating process. Time step was set to 0.1 ns in order not to lose any information. The position of the polarization axis within the particle was in axial direction.

## Data availability

Source data are provided with this paper.

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

## Acknowledgements

H.H. acknowledges the financial support by the Research Grants Council of the Hong Kong Special Administrative Region, China (Project No. PolyU152140/19E). H.Y. is grateful for postgraduate fellowship support from the Hong Kong Polytechnic University. D.L. acknowledges the financial support by National Natural Science Foundation of China through the Excellent Young Scientists Fund (Grant No. 62022001).

## Author contributions

H.H. and D.L. conceived the idea and designed the research. H.Y. and S.L. synthesized the material. H.Y. performed XRD, UV-Vis, UPS, SEM, BET, PL and pyro-catalytic experiments. Y.F. performed the COMSOL simulation. X.G. and Y.Z. performed morphology characterization. Z.L. draw graphic abstract. R.D. and J.H. performed PMF characterization. X.C. and H.-Y.T. help with nanosecond laser test. H.Z. and T.W.B.L. helped with the gas chromatography test. H.Y., D.L, C.-H.L., and H.H. performed data and theoretical analysis. H.Y. and H.H. wrote the paper with the help from all of the other co-authors.

## Competing interests

The authors declare no competing interests.
