## [Peer Review File · Nature Communications]

Accelerated Pyro-Catalytic Hydrogen Production Enabled by Plasmonic Local Heating of Au on Pyroelectric BaTiO₃ NanoparticlesREVIEWER COMMENTS

Reviewer #1 (Remarks to the Author):

What are the noteworthy results?

- The authors present a new approach of locally heating pyroelectric BaTiO₃ nanoparticles by plasmonic heating of surface grown Au, instead of heating / cooling the surrounding medium. The great advantage lies in the possible rapid yet efficiently heating and cooling of the pyroelectric particles. The application of this procedure in hydrogen production experiments showed a high hydrogen production rate of 181.2 μmol/gh under pulsed laser irradiation.

Will the work be of significance to the field and related fields? How does it compare to the established literature? If the work is not original, please provide relevant references.

- The work is of significance to the field of pyrocatalytic H₂ generation as well as pyrocatalytic pollutant degradation.
- The work presents an interesting approach to the problem of slow and energy-consuming thermal excitation of pyroelectric materials. Previous approaches excite pyroelectric catalysts almost exclusively indirectly via heating/cooling of the medium or, in a few cases, after immobilization or by means of IR radiation. The high excitation frequency achieved in this work is unique, even if locally limited.

Does the work support the conclusions and claims, or is additional evidence needed?

- The work basically supports all its conclusions and claims.
- One aspect that causes ambiguity is the postulated structure of the nanoparticles. Based on the SEM results, the morphology is described as "a morphology of three-dimensional hierarchically structured coral-like shape with an average size of several hundred nanometers". It is not clear whether the individual "dendrites" of this coral-like structure are intergrown/sintered or just loosely agglomerated. Later sections (PFM, simulations, calculations) always refer to elongated single particles. This ambiguity should be removed by appropriate evidence or clarification.

Are there any flaws in the data analysis, interpretation and conclusions? Do these prohibit publication or require revision?

- There are inconsistencies in the calculation of various parameters based on particle shape and size. While a particle length of 200 nm was used to calculate ΔV (Equation 3), a rectangular particle with dimensions of 50x50x1000 nm was chosen to calculate the total number of particles (Equation S5). For the thermal simulation W-BaTiO₃ a cylindrical particle ($l = 1 \mu\text{m}$, $r = 50 \text{ nm}$) was used. In addition, there are discrepancies of particle morphology between the SEM images (μm -sized agglomerates of X00 nm long particles) and the PFM measurements (single particles of about 400x75 nm). All calculations and interpretations should be unified before publication.
- Error bars are missing in the presented results for H₂ generation (e.g. fig. 3a-d). Furthermore, information on the number of experiments performed per experimental parameter set is missing. These experiments must always be performed several times (at least 2 times) to obtain a reliable result. This is particularly important due to the fact that a very small amount of catalyst, only 0.5 mg, was weighed in. Typical weighing errors of analytical balances are $\pm 0.1 \text{ mg}$, which would correspond to $\pm 20\%$ for this small sample quantity.
- Fig. 5 raises some questions. First, this figure is missing the effect of thermal excitation on the band structure of pyroelectric BaTiO₃. The pyroelectric effect should cause band bending (see Y. Liu, X. Wang, Y. Qiao, M. Min, L. Wang, H. Shan, Y. Ma, W. Hao, P. Tao, W. Shang, J. Wu, C. Song, T. Deng, Pyroelectric Synthesis of Metal-BaTiO₃ Hybrid Nanoparticles with Enhanced Pyrocatalytic Performance, ACS Sustainable Chemistry & Engineering 2018, 7, 2602-2609.). Furthermore, it is unclear where the

negative charges in the conduction band originate. In pyrocatalysis, no excitation of electrons from the valence band occurs without the use of UV light. In Liu et al., electrons are accumulated in the conduction band but only within the space region.

- For the calculation of the pyroelectric voltage (ΔV ; equation 3), T of 42 K was assumed. However, it is clear from Fig. 4b that a ΔT of 42 K is only achieved at the point (0,0,0) and not over the entire particle section P-BaTiO₃. However, in the calculation of ΔV , $l = 200$ nm is used as the length, thus ΔV is calculated for the entire P-BaTiO₃ section. It can be seen from Fig. 4c that ΔT for P-BaTiO₃ only reaches a maximum of about 4.2 K for a small period of time. This means that ΔV should only be in the range 0.19 - 0.29 V or less.
- The authors estimate the upper limit for pyro-catalytic hydrogen production using a laser with a repetition rate of 1000 Hz to be 500 mmol/g-1 h-1. They assume a H₂ production of 500 μ mol/g-1 h-1 at a repetition rate of the laser of 10 Hz. How do the authors explain the increase of the H₂ production rate by a factor of 1000 at a 100-fold increase of the amount of laser pulses?
- The possible maximum amount of generated hydrogen is calculated via total surface charge. However, there are already published models of pyrocatalytic hydrogen generation, that take several efficiency factors into account (i.e. J. Schlechtweg, S. Raufeisen, M. Stelter and P. Braeutigam, A novel model for pyro-electro-catalytic hydrogen production in pure water, Physical Chemistry Chemical Physics, 2019 / M. U. de Vivanco, M. Zschornak, H. Stoecker, S. Jachalke, E. Mehner, T. Leisegang and D. C. Meyer, Pyroelectrically-driven chemical reactions described by a novel thermodynamic cycle, Phys. Chem. Chem. Phys., 2020). For example surface charge will be "lost" by a certain degree by capacitive effects. This "loss" of surface charge should be taken into consideration by calculating H₂max.
- The authors assumed a global influence on the pyroelectric effect by locally heating the pyroelectric BaTiO₃ particles. In fig. 4 it is shown that the plasmonic heating of Au particles will only affect a small part of the BaTiO₃ particle. How do the authors explain such a global influence on the BaTiO₃ particle by this type of heating?

Is the methodology sound? Does the work meet the expected standards in your field?

- The methods used are sound and meet the expected standards.

Is there enough detail provided in the methods for the work to be reproduced?

- The methods used (Synthesis, Characterization, H₂ production experiments, simulation) are not sufficiently described to allow the work to be reproduced.
- Synthesis : In order to be able to synthesize the Au-decorated BaTiO₃ particles, information on the origin/manufacturer and properties of all chemicals used (e.g. TiO₂: particle size, crystal phase?) are crucial. Furthermore, a synthesis protocol for the preparation of the Au particles by citrate reduction method (amounts of chemicals and solvents, boiling time, etc.) is missing. How often the catalysts are washed with which solvent? Were the synthesis methods developed in-house or were there templates in the literature?
- Characterization : Important details about the measurement conditions of all used characterization methods are missing. For the XRD measurements, for example, the scan rate, the scan range and the wavelength of the XRD radiation used are missing. For the SEM measurements, information on sample preparation as well as on the adhesive pads used and any sample coating is missing. The same applies to most of the other methods. The authors should also support the assumption that BaTiO₃ is an indirect bandgap material with a reference. The authors did not describe how they estimated the tetragonality c/a from the split of the $\{200\}$ peaks in the XRD pattern.
- H₂ production experiments: Precise information on the reactor geometry and the sealing of this reactor is missing. Furthermore, it is unclear whether the extracted gas volume from the reactor is refilled with N₂ after sampling. In this case, this dilution of the H₂ concentration in the gas space of the reactor should be included in the evaluation. The determination of the H₂ content is insufficiently described. In gas chromatography, parameters such as type/length/diameter/material of the chromatographic construction must be specified. Furthermore, information on the exact detector type

as well as the calibration parameters (working range, limit of detection and limit of quantification, y-axis intercept and slope in a linear regression, etc.) are missing.

- Calculations : Some essential calculations are not comprehensible, because not all used values and/or boundary conditions are given. For example, the particle dimensions or surface area are missing in the calculation of Q (equation S3). These calculations are the basis for the prediction of the maximum possible pyrocatalytic H₂ production rate. The authors should define the position of the polarization axis within the particle.

Further remarks

- There are some minor spelling issues and sometimes the space between end of sentence and reference is missing.
- Sometimes the space between value and its unit is missing.
- ~ should be substituted by 'ca.' or 'about'. In some cases, like the produced hydrogen amount (181,2 μmol/gh), errors should be provided.
- Sometimes there are spaces before and after the '=', sometimes not (see SI (1) (2))
- Reference [7] E. Gutmann, A. Benke and K. Gerth et al., J. Phys. Chem. C, 2012,116, 5383-5393 should be replaced by S. Raufeisen, M. Stelter, P. Braeutigam, PLoS ONE 15(2): e0228644, since DCF method has been greatly improved there (pH-independent etc.)
- The authors write: „However, the currently available pyroelectric materials, whose pyro-catalytic capability relies on the variation of ambient temperature, show low pyro-catalytic efficiencies.“ Compared to photocatalysis, the catalytic activity of pyroelectric materials may be low, but there are publications in which heating up to 60/70°C is used. - For example: Raufeisen, M. Stelter, P. Braeutigam, PLoS ONE 15(2): e0228644, since DCF method has been greatly improved there (pH-independent etc.)
- The authors calculate in equation 1 the short-circuit pyro-current I for pyro-catalytic reactions. Here they use values for the pyroelectric coefficient (10~50 nCcm⁻²K⁻¹) and the environmental temperature ramping rate (0.1 K/s), which are not or not comprehensibly supported by literature references.
- The authors should provide some information about previous work and especially amounts of produced hydrogen via pyrocatalysis by other groups. This information would help to compare their approach with other approaches on pyrocatalytic hydrogen production and heating procedures.

Reviewer #2 (Remarks to the Author):

The work of You et al. reports on the fabrication, characterization and testing of a pyro-catalytic material based on BaTiO₃ nanoparticles modified with gold nanoparticles. The plasmonic properties of the Au NPs ensure rapid repeated heating of the catalyst in the direct environment of the particles upon irradiation with a (pulsed) laser source. The advantage of rapid temperature cycling enabled by plasmonic nanostructures under pulsed laser irradiation, is clearly evidenced in figure 3b and 3d, which is in my view the highlight of this work.

While pyro-catalytic is an original, less-studied field of science, I do not recommend this work for publication since the advance in state of the art is limited. The best performing material in the water splitting reaction, yields about 181 micromoles of H₂ per g catalyst per hour. Many reported photocatalysts easily exceed yields of 1 millimole of H₂ per g catalyst per hour, even when seawater is used as a feed (e.g. <https://doi.org/10.1039/C7EE02464A> (order of 3 mmol/g/h), and <https://doi.org/10.1002/cssc.201801819> (order of 4.5 mmol/g/h), amongst others). The authors claim that their system may reach a yield of 500 mmol/g/h, which would be far beyond state of the art, yet this purely hypothetically based on a theoretically optimized material. Without experimental validation of such claims, this should not be considered an actual advance.

In addition, the authors compare the performance of their catalysts with other photocatalytic materials, which is something I would like to question. While, indeed, light lies at the basis of the

reaction, the reaction itself is actually heat-driven, due to plasmonic heating. Hence, I wonder if it would not be more fair to benchmark the activity against other (photo)thermal catalysts, rather than purely light-driven materials.

Finally, it is not entirely clear from the description in the manuscript how local heating of a photo-active material (BaTiO₃), finally results in excitation of electron-hole pairs across the band gap, which is greater than the energy of incident light (cfr. figure 5). Since this is the primary assumption behind the entire process, I believe this should be explained and demonstrated in more detail.

As minor comments, I would like to add that the scale bar in fig 2e is supposedly wrong (10 nm rather than 100 nm), and the use of the term 'significantly accelerated' is not appropriate in the title.

Reviewer #3 (Remarks to the Author):

In this work, the authors used plasmonic local heating to effectively expedite the pyro-catalytic hydrogen production by Au/BaTiO₃ nanoparticles. Conventional pyro-catalysis is not very effective due to the lack a large and frequent temperature change. This is a novel and very interesting working that remove the obstacle in the potential application of pyro-catalysis. The authors have conducted systematic experimental and computer simulation work to understand the underlying mechanism. Overall the manuscript is well written and nicely illustrated. There are some minor issues need to be clarified before the manuscript can be accepted.

1. Surface area is quite important in the evaluation of catalytic performance, which should be equally important for pyro-catalysts used in the current work. BET test of the pyroelectric BaTiO₃ nanoparticles should be conducted to evaluate the surface area of pyroelectric BaTiO₃.
2. I'm wondering whether the high peak power of the laser irradiation would cause any damage to the catalyst after the pyro-catalysis experiment? It is better to provide XPS or TEM results of the catalyst after pyro-catalysis to evaluate the stability issue.
3. Hydrogen and oxygen are produced through pyro-catalysis, as shown in the supporting information. Is it possible to estimate the light-to-chemical energy conversion efficiency, so that researchers of different groups are able to compare their results?
4. The author wrote, "internal electric field built up from the surface pyroelectric charges can further facilitate the charge separation and charge transfer for pyro-catalytic hydrogen and oxygen production", but there is no electric field indicated in Fig. 5. I suggest the authors may include this effect in the schematic drawing.
5. According to the calculation, the H₂ production rate should much higher than the obtained results, what restrict reported pyro-catalytic results?
6. According to the results given in Fig. S7a (or c or e), the temperature of BTO should diminish after one-pulse illumination and eventually reach the ambient temperature, so that the surface integral of Eq. 3 should be zero. If so, it is unclear how the results in Fig. S7f was obtained.
7. The present calculations were performed based on a single AuNP induced pyroelectric charge accumulation. Since each BTO may be decorated by many AuNPs, a collective plasmonic heating effect may exist in the BTO and changes the heating process as discussed in Fig. S7f. The authors should clarify this issue.

REVIEWER COMMENTS

Reviewer #1 (Remarks to the Author):

Comment 1: What are the noteworthy results?

- The authors present a new approach of locally heating pyroelectric BaTiO₃ nanoparticles by plasmonic heating of surface grown Au, instead of heating/cooling the surrounding medium. The great advantage lies in the possible rapid yet efficiently heating and cooling of the pyroelectric particles. The application of this procedure in hydrogen production experiments showed a high hydrogen production rate of 181.2 μmol/g/h under pulsed laser irradiation.

Will the work be of significance to the field and related fields? How does it compare to the established literature? If the work is not original, please provide relevant references.

- The work is of significance to the field of pyro-catalytic H₂ generation as well as pyro-catalytic pollutant degradation.

- The work presents an interesting approach to the problem of slow and energy-consuming thermal excitation of pyroelectric materials. Previous approaches excite pyroelectric catalysts almost exclusively indirectly via heating/cooling of the medium or, in a few cases, after immobilization or by means of IR radiation. The high excitation frequency achieved in this work is unique, even if locally limited.

Reply: We thank the reviewer for the positive comment.

Comment 2: Does the work support the conclusions and claims, or is additional evidence needed?

- The work basically supports all its conclusions and claims.

- One aspect that causes ambiguity is the postulated structure of the nanoparticles. Based on the SEM results, the morphology is described as "a morphology of three-dimensional hierarchically structured coral-like shape with an average size of several hundred nanometers". It is not clear whether the individual "dendrites" of this coral-like structure are intergrown/sintered or just loosely agglomerated. Later sections (PFM, simulations, calculations) always refer to elongated single particles. This ambiguity should be removed by appropriate evidence or clarification.

Reply: The individual "dendrites" of coral-like structure BaTiO₃ is grown with many nanoparticles agglomerated along one dimension, as seen from the SEM images. For easy characterization of PFM, the coral-like shaped BaTiO₃ was shattered into separated particles by using ultrasonic vibration.

In our simulations and calculations, for simplicity, we treat the coral-like structure as an elongated single particle with a Au NP attached on it. This elongated single particle is the basic unit (or "unit cell") for simulation and calculation. We think this is a reasonable simplification and will not affect the main conclusions of our manuscript.

Comment 3: Are there any flaws in the data analysis, interpretation and conclusions? Do these prohibit publication or require revision?

• There are inconsistencies in the calculation of various parameters based on particle shape and size. While a particle length of 200 nm was used to calculate ΔV (Equation 3), a rectangular particle with dimensions of 50 x 50 x 1000 nm was chosen to calculate the total number of particles (Equation S5). For the thermal simulation, W-BaTiO₃ a cylindrical particle ($l = 1 \mu\text{m}$, $r = 50 \text{ nm}$) was used. In addition, there are discrepancies of particle morphology between the SEM images (μm -sized agglomerates of X00 nm long particles) and the PFM measurements (single particles of about 400 x 75 nm). All calculations and interpretations should be unified before publication.

Reply: In revised simulations and calculations, we unify the shape and size of the particle to be cylindrical ($l = 1 \mu\text{m}$ and $r = 50 \text{ nm}$). The cylindrical particle is used to simulate the “coral-like” structure. The length of the cylindrical particle is chosen to be 1 μm , which is long enough for heat dissipation. Too long a cylindrical particle will only add computation time while it does not help improve the computation accuracy too much.

Experimentally, SEM image (Fig.2a) shows that the BaTiO₃ particle size is within the range of 50-200 nm. In PFM characterization, the coral-like shaped BaTiO₃ was shattered into separated particles by using ultrasonic vibration. It can be seen from the PFM image (Fig.S1), the BaTiO₃ particles have a size of around 75 nm and are agglomerated into a chain of ~400 nm.

In a word, the experimental particle size varies. It is reasonable for us to choose a particle size of 100 nm (radius of 50 nm) for simulation and calculation.

Revision: Particle size and shape in the calculations was unified to be a cylindrical ($l = 1 \mu\text{m}$ and $r = 50 \text{ nm}$). The clarification “In PFM characterization, the coral-like shaped BaTiO₃ was shattered into separated particles by using ultrasonic vibration.” was added in Supporting Information.

Comment 4: • Error bars are missing in the presented results for H₂ generation (e.g. fig. 3a-d). Furthermore, information on the number of experiments performed per experimental parameter set is missing. These experiments must always be performed several times (at least 2 times) to obtain a reliable result. This is particularly important due to the fact that a very small amount of catalyst, only 0.5 mg, was weighed in. Typical weighing errors of analytical balances are $\pm 0.1 \text{ mg}$, which would correspond to $\pm 20\%$ for this small sample quantity.

Reply: For the weighing accuracy, we used an analytical balance with an accuracy of $\pm 0.01 \text{ mg}$ in the preparation of Au/BaTiO₃ nanoparticle suspension. Besides, the 0.5 mg catalyst (Au/BaTiO₃ nanoparticles) was taken from the uniform catalyst suspension of a larger volume by a pipette, not by weighing (see the revised SI for more information). Therefore, the amount of catalyst measured was reasonably accurate.

Revision: To convince the reviewer, we re-run the H₂ production experiments with larger amount of catalyst. In the revised pyro-catalytic experiment, around 2.62 mg (0.5 mg in the

original work) Au decorated BaTiO₃ nanoparticles were dispersed in 12 mL (4 mL in the original work) deionized water. The aqueous suspension was put into a 30 mL (10 mL in the original work) quartz reactor. The obtained H₂ production rate (133.1±4.4 μmol·g⁻¹ h⁻¹) (Fig. R1 below) is lower than the original work (181.2 μmol·g⁻¹ h⁻¹). We believe this is a reasonable result since the higher concentration of catalysts and larger reactor used will result in more light scattering. Besides, the original reported value is the best one among a group of experiments, while the current one is the averaged value.

Error bars and number of experiments performed per experimental parameter set are added in the revised manuscript and SI. The revised figure is also shown below as Fig. R1 (Fig. 3a-d in the revised manuscript).

Fig. R1 | H₂ generation from water splitting by Au/BaTiO₃ NPs. a, Hydrogen generation under different reaction time. **b**, Normalized H₂ production rate (normalized to the production rate of Au/BaTiO₃ NPs) of different samples under the irradiation by a 532 nm nanosecond laser. **c**, Hydrogen generation under the irradiation of a 532 nm nanosecond pulsed laser with different power densities. **d**, H₂ generation by Au/BaTiO₃ NPs illuminated under different light sources.

Comment 5: • Fig. 5 raises some questions. First, this figure is missing the effect of thermal excitation on the band structure of pyroelectric BaTiO₃. The pyroelectric effect should cause band bending (see Y. Liu, X. Wang, Y. Qiao, M. Min, L. Wang, H. Shan, Y. Ma, W. Hao, P. Tao, W. Shang, J. Wu, C. Song, T. Deng, Pyroelectric Synthesis of Metal-BaTiO₃ Hybrid Nanoparticles with Enhanced Pyrocatalytic Performance, ACS Sustainable Chemistry & Engineering 2018, 7, 2602-2609.). Furthermore, it is unclear where the negative charges in the conduction band originate. In photo-catalysis, no excitation of electrons from the valence band

occurs without the use of UV light. In Liu et al., electrons are accumulated in the conduction band but only within the space region.

Reply and revision: We agree with the reviewer that band tilting will occur under pyroelectrically induced internal electric field. We revise Fig.5 (also Fig. R2 below) in our revised manuscript to reflect the band tilting effect under thermal excitation.

The source of charges mainly comes from two origins: (1) free electrons in the conduction band, and (2) screen charges on the surface (Angew. Chem. Int. Ed. 2022, 61, e202110429). The free electrons may come from defects (such as oxygen vacancies) to keep neutrality of the material's internal charges. In the present case, the hot electrons injected from the Au nanoparticles to BaTiO₃ particles is another source of free electrons. Those free electrons will drift under the pyroelectrically induced internal field and migrate to the surface to participate in the catalytic reaction. The surface screen charges come from charged species in the liquid to compensate the polarization-induced surface bound charges. Under thermal fluctuation, the spontaneous polarization may vary and the redundant screen charges may participate in the catalytic reaction. At this stage, we cannot differentiate the contributions from free electrons and screen charges. The differentiation of these two types of charges requires delicate design of experiments and detailed understanding of different catalytic reaction pathways. To shed light to future study on this tough mechanism, we add the above discussion in revised SI and supplement with a schematic drawing of pyro-catalysis mechanism (Fig.S12 in revised SI and also Fig. R3 below).

Fig. R2 | Schematic illustration of pyro-catalytic hydrogen generation of Au/BaTiO₃ NP driven by surface plasmon local heating.

Fig. R3 | Mechanism of pyro-catalysis

Comment 6: • For the calculation of the pyroelectric voltage (ΔV ; equation 3), T of 42 K was assumed. However, it is clear from Fig. 4b that a ΔT of 42 K is only achieved at the point (0,0,0) and not over the entire particle section P-BaTiO₃. However, in the calculation of ΔV , $l = 200$ nm is used as the length, thus ΔV is calculated for the entire P-BaTiO₃ section. It can be seen from Fig. 4c that ΔT for P-BaTiO₃ only reaches a maximum of about 4.2 K for a small period of time. This means that ΔV should only be in the range 0.19 - 0.29 V or less.

Reply and revision: It is true that neither the point temperature change (around 42K at point (0, 0, 0)) nor the domain averaged temperature change (around 4.2K of the domain P-BaTiO₃) can be directly used to evaluate the overall voltages induced by the pyroelectric and thermoelectric effects during one laser pulse illumination. In order to give a more precise comparison between the two effects, we calculate the available charges (electrons) induced by these two effects. The reason behind this is that the amount of available charges induced by pyroelectric and thermoelectric effects is more relevant to catalysis than the induced voltage. To calculate the thermoelectrically induced charges, we set two circular cut planes, marked as **P_c** and **P_e** in Fig. R4a. The temperature difference between the two planes (**P_c** and **P_e**) will induce current flows due to thermoelectric effect in the axial direction of the cylindrical particle, as marked by red arrows. The length of red arrow schematically shows the magnitude of current density (not drawn to scale, Fig. R4a left panel). Similarly, we also consider the current flows in a direction perpendicular to the axial direction, as marked by blue arrows in Fig. R4b left panel. Two cut planes parallel to the xy -plane are created within the cylindrical BaTiO₃ particle. By drawing tangent lines from the center of the Au nanosphere to the BaTiO₃ surface (Fig. R4b, right panel), the plane **P_t** is formed by all the points of tangency. Simple calculation shows that **P_t** is 7.98 nm from the top of the cylindrical BaTiO₃ particle. For symmetry consideration, the other cut plane **P_b** is parallel to **P_t** but at an equal distance of 7.98 nm above the bottom of BaTiO₃ particle, serving as a reference plane for calculating the temperature difference. These

two planes are reasonably chosen to evaluate the thermoelectric effect in z -direction since the BaTiO₃ surface above P_t (highlighted by red colored zone in Fig. R4b) is regarded as being directly heated by the Au nanosphere and thus be treated as the heat source in the thermal process. The surface-averaged temperature in the P_c and P_e cut planes are given in Fig. R4c. And the surface-averaged temperatures in the P_t and P_b cut planes are given in Fig. R4d. It can be seen that, the temperature difference in axial direction is much higher as P_c and P_e cut planes are far apart. On the contrary, the surface-averaged temperature difference in P_t and P_b cut planes is much smaller due to their smaller separation distance (84.04 nm).

Fig. R4 | Structural model for simulation: **a**, One faceted Au nanosphere ($h=0.5$ nm) suspending over a cylindrical BaTiO₃ particle (water gap=1nm). Left panel: cut planes used to calculate the temperature difference in axial direction (their positions are represented by black lines and marked as P_c and P_e). Right panel: side-view of P_c and P_e cut planes. **b**, The same schematic as that shown in **a**, but with two different cut planes, i.e., P_t and P_b , used to calculate the temperature difference in z -direction. **c**, Surface-averaged temperature of P_c (blue curve, marked as "center") and P_e (red curve, marked as "extremity"). **d**, Surface-averaged temperature of P_t (blue curve, marked as "top") and P_b (red curve, marked as "bottom").

The quantity of electrons induced by thermoelectric effect can be derived as follows [Nanophotonics, 2018, 7, 1917-1927]:

$$Q = \int I dt = \int \sigma(T) E dA dt = \int \sigma(T) \frac{V}{l} dA dt = \int S \frac{\sigma(T) \Delta T}{l} dA dt \quad (R1)$$

where I , σ , E , V , l , A , t , S , and ΔT are the current, the electrical conductivity [J. Phys. Colloques, 1972, 33, C2-120-C2-122], electric field, thermoelectric voltage, the distance between two cut planes, surface area, time, Seebeck coefficient and the temperature difference between two end planes, respectively. Fig. R5a and Fig. R5b below show the quantities of electrons induced thermoelectric effect due to the temperature difference in axial direction and z -direction, respectively.

Fig. R5 | Quantity of electrons induced by thermoelectric effect in **a**, axial direction; and **b**, z -direction.

The quantity of electrons induced by pyroelectric effect can be expressed by Eq. R2:

$$Q = p \cdot \int \Delta T (\hat{n} \cdot \hat{P}) dA \quad (R2)$$

where p is the pyroelectric coefficient with a value between 20 and 30 $\text{nC} \cdot \text{cm}^{-2} \cdot \text{K}^{-1}$, ΔT is the temperature rise in heating (or drop during cooling), \hat{n} is the unit vector along the surface normal direction, and \hat{P} is the unit vector along spontaneous polarization direction.

Without losing generality, we compare the cases for polarizations along the axial direction and z -direction. The integral of $\Delta T (\hat{n} \cdot \hat{P}) dA$ over the upper surface (corresponding polarization

along z -axis) of the cylindrical particle can be up to $1.71 \times 10^{-13} \text{ m}^2\text{K}$ (Fig. R6), which is much larger than the integral over the \mathbf{P}_c cut plane (corresponding polarization along axial direction) (location of \mathbf{P}_c can be found from Fig. R4a). Moreover, the pyroelectric charges generated on the \mathbf{P}_c cut plane are difficult to diffuse to the surface for catalytic reaction due to the low conductivity of BaTiO_3 . Therefore, we consider only the polarization along z -axis for the estimation of the upper limit of H_2 production.

Taking the pyroelectric coefficient (p) of $30 \text{ nC} \cdot \text{cm}^{-2} \cdot \text{K}^{-1}$ as an example, we can easily calculate that the pyroelectric charges are $5.12 \times 10^{-17} \text{ C}$, which are much larger than those due to thermoelectric effect (Fig.R5). So the thermoelectric effect induced catalysis can be neglected in the present case.

The results and corresponding discussion are added in the revised manuscript and Supporting Information.

Fig. R6 | Area integrated temperature change ($\int \Delta T(\hat{n} \cdot \hat{P})dA$) over the upper half surface of the cylindrical BaTiO_3 particle after one laser pulse irradiation on a 9-nm Au NP. Inset shows the integrated area (shaded part) of the particle.

Comment 7: • The authors estimate the upper limit for pyro-catalytic hydrogen production using a laser with a repetition rate of 1000 Hz to be $500 \text{ mmol} \cdot \text{g}^{-1} \cdot \text{h}^{-1}$. They assume a H_2 production of $500 \text{ } \mu\text{mol} \cdot \text{g}^{-1} \cdot \text{h}^{-1}$ at a repetition rate of the laser of 10 Hz. How do the authors explain the increase of the H_2 production rate by a factor of 1000 at a 100-fold increase of the amount of laser pulses?

Reply: In our current calculation, each cylindrical BaTiO_3 NP with a length of $1 \text{ } \mu\text{m}$ is assumed to be locally heated by one Au NP. This is not efficient since appreciable amount of temperature change can only be found in the region of less than 50 nm away from the heat source of Au NP (Fig. S10e in the revised SI). If the length of BaTiO_3 NP can be shortened by one tenth down

to 100 nm (this is experimentally feasible), then the H₂ production rate will be increased by 10 times. On the other hand, if we can increase the repetition rate of our laser from the current 10 Hz to 1000 Hz (laser product available in the market), the H₂ production rate will be further increased by 100 times within a unit time. In total, we expect a reasonable increase by a factor of 1000.

Comment 8: • The possible maximum amount of generated hydrogen is calculated via total surface charge. However, there are already published models of pyro-catalytic hydrogen generation, that take several efficiency factors into account (i.e. J. Schlechtweg, S. Raufeisen, M. Stelter and P. Braeutigam, A novel model for pyro-electro-catalytic hydrogen production in pure water, *Physical Chemistry Chemical Physics*, 2019 / M. U. de Vivanco, M. Zschornak, H. Stoecker, S. Jachalke, E. Mehner, T. Leisegang and D. C. Meyer, Pyroelectrically-driven chemical reactions described by a novel thermodynamic cycle, *Phys. Chem. Chem. Phys.*, 2020). For example, surface charge will be “lost” by a certain degree by capacitive effects. This “loss” of surface charge should be taken into consideration by calculating H₂ max.

Reply: It is true that there are certain efficiency loss factors that may restrict the maximum achievable pyro-catalytic H₂ production rate. Those factors include surface charge loss by capacitive effects, insufficient absorption of the incident light, and charge recombination, etc. [J. Schlechtweg, *et al.*, *Phys. Chem. Chem. Phys.*, 2019, 21, 23009-23016; M. U. de Vivanco *et al.*, *Phys. Chem. Chem. Phys.*, 2020, 22, 17781-17790]. What we calculated here is just a theoretical upper limit of the pyro-catalytic H₂ production rate. A more comprehensive theoretical model to include the above loss factors is not the focus of the current work and will be done in the future.

Revision: We add the above discussion on the possible loss factors in the revised manuscript.

Comment 9: • The authors assumed a global influence on the pyroelectric effect by locally heating the pyroelectric BaTiO₃ particles. In Fig. 4, it is shown that the plasmonic heating of Au particles will only affect a small part of the BaTiO₃ particle. How do the authors explain such a global influence on the BaTiO₃ particle by this type of heating?

Reply: It is not necessary to have a global heating in order to get the pyroelectric charges. Since pyroelectric charges for catalytic reaction are on the surface, a local heating on one end of the surface will be able to get some pyroelectric charges even if the other end of surface is not heated. The difference is that, for a global heating, during the cooling process, the other end of the surface will release equal amount of pyroelectric charges of the same sign, as we schematically shown in Fig. R3 (Fig.S12 in revised SI). As compared with global heating, local heating will generate less pyroelectric charges. However, the main conclusion remains unchanged.

Revision: Our original calculation of pyroelectric charges by assuming a global heating is just a very rough estimation. To be more precise, in this revised manuscript, by taking the local

heating effect into consideration, we use the integration as shown in Eq. R2 (please refer to our reply to **Comment 6** above). The area integrated temperature change ($\int \Delta T(\hat{n} \cdot \hat{P})dA$) results are shown in Fig.R6 (Fig.S15, revised SI), which can be used for further calculation of pyroelectric charges.

Comment 10: Is the methodology sound? Does the work meet the expected standards in your field?

- The methods used are sound and meet the expected standards.

Reply: Thanks.

Comment 11: Is there enough detail provided in the methods for the work to be reproduced?

- The methods used (Synthesis, Characterization, H₂ production experiments, simulation) are not sufficiently described to allow the work to be reproduced.
- Synthesis : In order to be able to synthesize the Au-decorated BaTiO₃ particles, information on the origin/manufacturer and properties of all chemicals used (e.g. TiO₂: particle size, crystal phase?) are crucial. Furthermore, a synthesis protocol for the preparation of the Au particles by citrate reduction method (amounts of chemicals and solvents, boiling time, etc.) is missing. How often the catalysts are washed with which solvent? Were the synthesis methods developed in-house or were there templates in the literature?

Reply and revision: Thanks for your suggestion. The synthesis method was revised in the SI with more experimental details. The revised parts are highlighted in yellow. BaTiO₃ NPs were synthesized through a hydrothermal method with modifications. The synthesis of the Au/BaTiO₃ nanoparticles was developed in-house.

Comment 12: • Characterization: Important details about the measurement conditions of all used characterization methods are missing. For the XRD measurements, for example, the scan rate, the scan range and the wavelength of the XRD radiation used are missing. For the SEM measurements, information on sample preparation as well as on the adhesive pads used and any sample coating is missing. The same applies to most of the other methods. The authors should also support the assumption that BaTiO₃ is an indirect bandgap material with a reference. The authors did not describe how they estimated the tetragonality c/a from the split of the {200} peaks in the XRD pattern.

Reply and revision: Characterization methods were revised in the SI with revised parts highlighted in yellow.

BaTiO₃ is an indirect bandgap material (*J. Phys. Chem. C*, 2011, 115, 24373-24380; *Int. J. Mod. Phys. B*, 33, 2019,1950231; *Optik*, 2020, 211,164611.). The references were added in the revised manuscript.

The determination of tetragonality c/a is based on Bragg's law. The detailed calculation process is added in the revised SI.

Comment 13: • H₂ production experiments: Precise information on the reactor geometry and the sealing of this reactor is missing. Furthermore, it is unclear whether the extracted gas volume from the reactor is refilled with N₂ after sampling. In this case, this dilution of the H₂ concentration in the gas space of the reactor should be included in the evaluation. The determination of the H₂ content is insufficiently described. In gas chromatography, parameters such as type/length/diameter/material of the chromatographic construction must be specified. Furthermore, information on the exact detector type as well as the calibration parameters (working range, limit of detection and limit of quantification, y-axis intercept and slope in a linear regression, etc.) are missing.

Reply and revision: As suggested by the reviewer, more details about H₂ production experiments were added in the revised SI. Particularly, the aqueous suspension was put into a 30 mL pear-shaped quartz reactor and sealed using septa in advance, which was then evacuated and purged by N₂ for about 20 min to completely remove air. To detect the amount of hydrogen production, 300 μL gas component within the reactor was intermittently extracted and injected into a gas chromatograph (Agilent 7890B) with a thermal conductivity detector (chromatographic column: Agilent, length of 30 m and diameter 0.32 mm). 300 μL N₂ gas is refilled into reactor after sampling, which was taken into consideration in the calculation. The amount of hydrogen gas produced was calculated using a calibration curve ($y=137.78x+18.95$, $R^2=0.99962$) of hydrogen concentration versus peak area (working range: 300 ppm to 15000 ppm; limit of detection: 50 ppm; limit of quantification: 150 ppm).

Comment 14: • Calculations: Some essential calculations are not comprehensible, because not all used values and/or boundary conditions are given. For example, the particle dimensions or surface area are missing in the calculation of Q (equation S3). These calculations are the basis for the prediction of the maximum possible pyro-catalytic H₂ production rate. The authors should define the position of the polarization axis within the particle.

Reply and revision: We add more simulation details in the section of “**Simulation details**” of the revised SI with revised parts highlighted in yellow.

Without losing generality, we compare the cases for polarizations along the axial direction and z-direction. In Eq. R2 above, the integral of $\Delta T(\hat{n} \cdot \hat{P})dA$ over the upper surface (corresponding polarization along z-axis) of the cylindrical particle can be up to 1.71×10^{-13} m²K (Fig. R6), which is much larger than the integral over the **P_c** cut plane (corresponding polarization along axial direction) (location of **P_c** can be found from Fig. R4a). Moreover, the pyroelectric charges generated on the **P_c** cut plane are difficult to diffuse to the surface for catalytic reaction due to the low conductivity of BaTiO₃. Therefore, we consider only the polarization along z-axis for the estimation of the upper limit of H₂ production. We add the discussion in revised SI.

Comment 15: Further remarks

- There are some minor spelling issues and sometimes the space between end of sentence and reference is missing.
- Sometimes the space between value and it's unit is missing.
- ~ should be substituted by 'ca.' or 'about'. In some cases, like the produced hydrogen amount (181,2 $\mu\text{mol/gh}$), errors should be provided.
- Sometimes there are spaces before and after the '=', sometimes not (see SI (1) \square (2))
- Reference [7] E. Gutmann, A. Benke and K. Gerth et al., *J. Phys. Chem. C*, 2012,116, 5383-5393 should be replaced by S. Raufeisen, M. Stelter, P. Braeutigam, *PLoS ONE* 15(2): e0228644, since DCF method has been greatly improved there (pH-independent etc.)

Reply and revision: Thanks for your suggestion. We carefully revise those typos and mistakes in the revised manuscript. The suggested reference was also revised in the revised manuscript.

Comment 16:• The authors write: However, the currently available pyroelectric materials, whose pyro-catalytic capability relies on the variation of ambient temperature, show low pyro-catalytic efficiencies.” Compared to photocatalysis, the catalytic activity of pyroelectric materials may be low, but there are publications in which heating up to 60/70°C is used. - For example: S.Raufeisen, M. Stelter, P. Braeutigam, *PLoS ONE* 15(2): e0228644, since DCF method has been greatly improved there (pH-independent etc.)

Reply: We just said that the efficiency is low. We didn't imply that the heating temperature cannot go up to 60/70 °C. It is true that by simply increasing the temperature change (ΔT), one can get higher H_2 production for one thermal cycle. This is not enough. For application purpose, we need to have as many as possible thermal cycles within unit time. That is the problem we want to solve in this work since conventional heating cannot create multiple thermal cycles within a short time period.

Comment 17:• The authors calculate in equation 1 the short-circuit pyro-current I for pyro-catalytic reactions. Here they use values for the pyroelectric coefficient ($10\sim 50 \text{ nC cm}^{-2} \text{ K}^{-1}$) and the environmental temperature ramping rate (0.1 K/s), which are not or not comprehensibly supported by literature references.

Reply: The pyroelectric coefficient ($10\sim 50 \text{ nC cm}^{-2} \text{ K}^{-1}$) is supported by Ref 11 in our original manuscript. The references (J. Overgaard *et al.*, *J. Therm. Biol.* 36 2011, 409-416; J.S. Terblanche *et al.*, *Proc. R. Soc. B—Biol. Sci.* 2007, 274, 2935-2942; H.A. MacMillan and B.J. Sinclair, *J. Insect Physiol.* 2011, 57, 12-20.) that support the environmental temperature ramping rate (0.1 K/s) are added in the revised manuscript.

Comment 18:• The authors should provide some information about previous work and especially amounts of produced hydrogen via pyro-catalysis by other groups. This information would help to compare their approach with other approaches on pyro-catalytic hydrogen

production and heating procedures.

Reply and revision: We add the following table in the revised SI to compare the data from different groups.

Table R2 Hydrogen production via pyro-catalysis

Pyroelectric catalysts	Cold-hot cycle condition	Catalyst dosage (sacrificial agent/concentration)	Catalytic applications	Catalytic efficiency/rate constant	Ref.
BST nanoparticles	298K-323K, 10 min per thermal cycle	10 mg catalyst/10 mL H ₂ O (20 vol% methanol)	H ₂ production	47 $\mu\text{mol}\cdot\text{g}^{-1}$ (36 thermal cycles)	
2D few layers BP	288K-338K, 10 min per thermal cycle	1 mg catalyst/10 mL H ₂ O (20 vol% methanol)	H ₂ production	540 $\mu\text{mol}\cdot\text{g}^{-1}$ (24 thermal cycles)	
BaTiO ₃ single crystals	303-343K, 2 min per thermal cycle	3.1 g catalyst (without sacrificial agent)	H ₂ production	300 vol.-ppb	
PZT sheet	316K-319K	4.85 g, thickness of 170 μm , surface area of 49 cm^2 (without sacrificial agent)	H ₂ production	0.654 $\mu\text{mol}/\text{h}$	
SiC	300 K-333 K, 20 min per thermal cycle	50 mg catalyst/20 mL H ₂ O (10 vol% methanol)	H ₂ production	32.84 $\mu\text{mol}/\text{g}$	
Au/BaTiO ₃ nanoparticles	500 mW nanosecond laser/10 Hz	2.62 mg catalyst/12 mL H ₂ O (without sacrificial agent)	H ₂ production	133.1 \pm 4.3716 $\mu\text{mol}\cdot\text{g}^{-1}\cdot\text{h}^{-1}$	This work

Reviewer #2 (Remarks to the Author):

Comment 1: The work of You *et al.* reports on the fabrication, characterization and testing of a pyro-catalytic material based on BaTiO₃ nanoparticles modified with gold nanoparticles. The plasmonic properties of the Au NPs ensure rapid repeated heating of the catalyst in the direct environment of the particles upon irradiation with a (pulsed) laser source. The advantage of rapid temperature cycling enabled by plasmonic nanostructures under pulsed laser irradiation, is clearly evidenced in figure 3b and 3d, which is in my view the highlight of this work.

Reply: Thanks.

Comment 2: While pyro-catalytic is an original, less-studied field of science, I do not recommend this work for publication since the advance in state of the art is limited. The best performing material in the water splitting reaction, yields about 181 micromoles of H₂ per g catalyst per hour. Many reported photocatalysts easily exceed yields of 1 millimole of H₂ per g catalyst per hour, even when seawater is used as a feed (e.g. <https://doi.org/10.1039/C7EE02464A> (order of 3 mmol/g/h), and <https://doi.org/10.1002/cssc.201801819> (order of 4.5 mmol/g/h), amongst others). The authors claim that their system may reach a yield of 500 mmol/g/h, which would be far beyond state of the art, yet this purely hypothetically based on a theoretically optimized material. Without experimental validation of such claims, this should not be considered an actual advance.

Reply: We cannot agree with the reviewer at this point. Even for the state-of-the-art photocatalyst, it is not achieved through a single piece of work. Numerous researchers have made step-by-step contributions to this field to achieve the current state-of-the-art hydrogen production through photocatalytic water splitting. Although the performance of our pyro-catalyst cannot compete with that of the best photocatalysts at the current stage, it already surpasses the performance of all the pyro-catalysts heated by a conventional way.

Pyro-catalysis represents a mechanism different from that of photocatalysis. The key to having a high pyro-catalysis efficiency is to have as many as possible thermal cycles within a unit time. Conventionally, pyro-catalysis is done by heating/cooling the catalysts together with the surrounding water, which is very slow. We developed this local heating strategy to greatly enhance the number of heating/cooling cycles to 10 cycles per second, which is impossible to achieve through conventional heating/cooling.

The theoretical model we used to predict the upper limit of H₂ production rate is based on realistic materials parameters and feasible experimental set-up. It is not a groundless hypothesis. We believe that with the joint efforts of the research community, the pyro-catalytic H₂ production rate can reach an attractive value.

Moreover, apart from H₂ production, hydroxyl radicals were also detected in our experiment (Fig.S16, revised SI). This localized heating and generation of hydroxyl radicals is attractive for biological applications such as accurate treatment in tumor cells in nanoscale range without side effects. The wavelength of the excitation light can also be adjusted to near infrared band

(biological window) through changing the morphology of Au NPs, for instance, gold nanorods.

Comment 3: In addition, the authors compare the performance of their catalysts with other photocatalytic materials, which is something I would like to question. While, indeed, light lies at the basis of the reaction, the reaction itself is actually heat-driven, due to plasmonic heating. Hence, I wonder if it would not be more fair to benchmark the activity against other (photo) thermal catalysts, rather than purely light-driven materials.

Reply and revision: In our manuscript, the comparison of the performance of our synthesized catalyst in pyro-catalysis and photo-catalysis (Fig. 3d) is to rule out the photocatalytic contribution in our experiment. To avoid confusion, we revise the expression “an attractive value surpassing the best available photocatalyst” in the revised manuscript. We compare our pyro-catalytic result with those of the recently reported photo-thermal catalysts below.

Table R3 | Hydrogen production via photo-thermal catalysis

Photo-thermal catalysts	condition	Catalyst dosage (sacrificial agent/concentration)	Catalytic applications	Catalytic efficiency/rate constant	Ref.
Ni ₂ P/TiO ₂ nanocomposites	300 W Xenon lamp	14 mg catalyst/140 mL H ₂ O (20 vol% methanol)	H ₂ production	92.47 μmol·g ⁻¹ ·h ⁻¹	
Ag@SiO ₂ @TiO ₂ NS+Au NPs	100 mW cm ⁻² Xenon with 400 nm long-pass filter and UV LED (365 nm) with light intensity of 150·mW·cm ⁻²	5 mg catalyst/10mL H ₂ O (20 wt% glycerol)	H ₂ production	30.2 mmol·g ⁻¹ ·h ⁻¹	
Cu/TiO ₂	300 W full spectrum Xenon lamp and 450 W R-NIR light	10 mg catalyst/80 mL H ₂ O (20 vol% methanol)	H ₂ production	8.12 mmol·g ⁻¹ ·h ⁻¹	
Cu@Ni/g-C ₃ N ₄ composites	300 W Xenon lamp with filter of λ>420 nm	20 mg catalyst/80 mL H ₂ O (20 vol% triethanolamine)	H ₂ production	55 μmol·g ⁻¹ ·h ⁻¹	
Cu-TiO ₂	300 W Xenon lamp with an IR cutoff filter; 573.15K	50 mg catalyst/50mL H ₂ O (without sacrificial agent)	H ₂ production	18.09 μmol·g ⁻¹ ·h ⁻¹	
Au/BaTiO ₃ nanoparticles	500 mW nanosecond laser/10 Hz	2.62 mg catalyst/12 mL H ₂ O (without sacrificial agent)	H ₂ production	133.1±4.3716 μmol·g ⁻¹ ·h ⁻¹	This work

It can be seen from the above table, the hydrogen production rate of photo-thermal catalysis can be up to 30.2 mmol·g⁻¹·h⁻¹ (R. Song *et al.*, *Ind. Eng. Chem. Res.* 2018, 57, 7846-7854; S. Ng *et al.*, *Adv. Funct. Mater.* 2021, 31, 2104750; R. Song *et al.*, *AIChE J.* 2020, 66, e17008; X. Guo *et al.*, *ChemCatChem*, 2020, 12, 2745-2751; X. Zhang *et al.*, *ACS Appl. Energy Mater.*

2022, 5, 4564-4576). We want to emphasize that the mechanism of photo-thermal catalysis is different from the pyro-catalysis reported in this work and direct comparison of performance is not that meaningful. Photo-thermal catalyst can absorb UV-VIS light to perform photo-catalysis. At the same time, photo-thermal catalysis can also absorb NIR light or IR light to heat the whole catalyst to reduce the energy barrier of the rate-determining step, thus promoting the water splitting reaction. In our work, plasmonic effect induced local heating was used as the heat source to heat the pyroelectric material. This rapid heating and cooling of the pyroelectric particles induces pyro-catalysis.

Comment 4: Finally, it is not entirely clear from the description in the manuscript how local heating of a photo-active material (BaTiO_3), finally results in excitation of electron-hole pairs across the band gap, which is greater than the energy of incident light (cfr. figure 5). Since this is the primary assumption behind the entire process, I believe this should be explained and demonstrated in more detail.

Reply and revision: The mechanism of pyro-catalysis is different from that of photocatalysis. The source of charges for the catalytic reaction mainly comes from two origins: (1) free electrons in the conduction band, and (2) screen charges on the surface (Angew. Chem. Int. Ed. 2022, 61, e202110429).

The free electrons may come from defects (such as oxygen vacancies) to keep neutrality of the material's internal charges. In the present case, the hot electrons injected from the Au nanoparticles to BaTiO_3 particles is another source of free electrons. Those free electrons will drift under the pyroelectrically induced internal field and migrate to the surface to participate in the catalytic reaction.

The surface screen charges come from charged species in the liquid to compensate the polarization-induced surface bound charges. Under thermal fluctuation, the spontaneous polarization may vary and the redundant screen charges may participate in the catalytic reaction. At this stage, we cannot differentiate the contributions from free electrons and screen charges. The differentiation of these two types of charges requires delicate design of experiments and detailed understanding of different catalytic reaction pathways. To shed light to future study on this tough mechanism, we add the above discussion in revised SI and supplement with a schematic drawing of pyro-catalysis mechanism (Fig.S12 in revised SI).

Comment 5: As minor comments, I would like to add that the scale bar in fig 2e is supposedly wrong (10 nm rather than 100 nm), and the use of the term 'significantly accelerated' is not appropriate in the title.

Reply and revision: The scale bar in Fig. 2e is revised. We delete 'significantly' in the title.

Reviewer #3 (Remarks to the Author):

Comment 1: In this work, the authors used plasmonic local heating to effectively expedite the pyro-catalytic hydrogen production by Au/BaTiO₃ nanoparticles. Conventional pyro-catalysis is not very effective due to the lack a large and frequent temperature change. This is a novel and very interesting working that remove the obstacle in the potential application of pyro-catalysis. The authors have conducted systematic experimental and computer simulation work to understand the underlying mechanism. Overall, the manuscript is well written and nicely illustrated. There are some minor issues need to be clarified before the manuscript can be accepted.

Reply: Thanks.

Comment 2: Surface area is quite important in the evaluation of catalytic performance, which should be equally important for pyro-catalysts used in the current work. BET test of the pyroelectric BaTiO₃ nanoparticles should be conducted to evaluate the surface area of pyroelectric BaTiO₃.

Reply and revision: We conduct the BET test and add the results and corresponding discussion in the revised SI. As shown below, the surface area of catalyst is around 38.85 m²/g.

Fig. R7 | Surface area characterization of BaTiO₃ NPs.

Comment 3: I'm wondering whether the high peak power of the laser irradiation would cause any damage to the catalyst after the pyro-catalysis experiment? It is better to provide XPS or TEM results of the catalyst after pyro-catalysis to evaluate the stability issue.

Reply and revision: The morphology characterizations of Au/BaTiO₃ after pyro-catalysis are conducted and the results and corresponding discussion are added in the revised manuscript and SI. As shown in Figure R8 and R9 (Fig. S7 and S8 in the revised SI) below, after 90-min's pyro-catalysis, there is nearly no damage on Au/BaTiO₃ NPs, indicating that the Au/BaTiO₃ NPs

have good stability toward pyro-catalytic hydrogen production.

Fig. R8 | TEM image of Au/BaTiO₃ NPs after pyro-catalysis.

Fig. R9 | Morphology characterizations of Au/BaTiO₃ NPs after pyro-catalysis. a, b and c, HRTEM image of Au/BaTiO₃ NPs after pyro-catalysis with different radii of decorated Au NPs. d, HAADF-STEM image and corresponding elemental mapping of Ba, Ti, O, and Au elements of b.

Comment 4: Hydrogen and oxygen are produced through pyro-catalysis, as shown in the supporting information. Is it possible to estimate the light-to-chemical energy conversion efficiency, so that researchers of different groups are able to compare their results?

Reply and revision: The light-to-chemical energy conversion efficiency is calculated below and the results and corresponding discussion are added in the revised SI.

The value of the output chemical energy due to pyro-catalysis can be simply calculated as,

$$E_{\text{chem}} = 2n \cdot V_r \cdot N_A \cdot e$$

where n = mole number of hydrogen produced; V_t = threshold voltage (1.23 V) for water splitting; N_A = Avogadro's number; e = elemental charge.

According to the above equation, the chemical energy output due to pyro-catalysis per second in the present case is:

$$E_{\text{chem}} = 2n \cdot V_t \cdot N_A \cdot e = 2 \times 9.70 \times 10^{-11} \times 1.23 \times 6.02 \times 10^{23} \times 1.602 \times 10^{-19} \text{ (J} \cdot \text{s}^{-1}\text{)}$$
$$= 2.30 \times 10^{-5} \text{ (J} \cdot \text{s}^{-1}\text{)} = 0.023 \text{ mW}$$

The light-to-chemical energy conversion efficiency can be calculated as,

$$\eta = E_{\text{chem}}/E_{\text{input}} = 0.023 \text{ mW}/500 \text{ mW} = 0.0046\%$$

This value is quite low since the catalyst used is dispersed in the liquid and the majority of illuminated part is actually water, not the catalyst.

Comment 5: The author wrote, "internal electric field built up from the surface pyroelectric charges can further facilitate the charge separation and charge transfer for pyro-catalytic hydrogen and oxygen production", but there is no electric field indicated in Fig. 5. I suggest the authors may include this effect in the schematic drawing.

Reply and revision: The band tilting induced by internal electric field is added in Fig. 5 (as shown in Fig. R10 also). The figure and corresponding discussion are revised in the manuscript.

Fig. R10 | Schematic illustration of pyro-catalytic hydrogen generation of Au/BaTiO₃ NP driven by surface plasmon local heating.

Comment 6: According to the calculation, the H₂ production rate should much higher than the obtained results, what restrict reported pyro-catalytic results?

Reply and revision: There are several factors which may restrict the maximum achievable pyro-catalytic hydrogen production rate, for example, surface charge loss by capacitive effects,

insufficient absorption of the incident light, and charge recombination, etc. We add the corresponding discussion in the revised manuscript.

Comment 7: According to the results given in Fig. S7a (or c or e), the temperature of BaTiO₃ should diminish after one-pulse illumination and eventually reach the ambient temperature, so that the surface integral of Eq. 3 should be zero. If so, it is unclear how the results in Fig. S7f was obtained.

Reply: We agree with the reviewer's comment that the temperature of the BaTiO₃ will return to the ambient temperature after one-pulse illumination, resulting in opposite movement of electrons (and holes) during temperature rise and the subsequent temperature drop as shown in Fig. S10a (revised SI).

In the original manuscript, we used $\Delta V = \frac{l}{\epsilon_{33}^X \cdot \epsilon_0} \cdot p \cdot \Delta T$ (Eq.3) to calculate the pyroelectric voltage. The temperature change ΔT refers to the temperature rise during heating process, which is non-zero.

In this revised manuscript, we use $Q = p \cdot \int \Delta T(\hat{n} \cdot \hat{P})dA$ (Eq. R2) to calculate the pyroelectric charges. Similarly, the temperature change (ΔT) here only accounts for the temperature rise during heating process, while the integration runs over every specific point on the BaTiO₃ surface. This can be easily achieved by inserting one judging condition in Domain ODE and DAE Interface (dode) in Comsol.

Revision: The sentence was revised in the Supporting Information to emphasize the calculation was only in the heating process of one heating/cooling cycle.

Comment 8: The present calculations were performed based on a single Au NP induced pyroelectric charge accumulation. Since each BaTiO₃ may be decorated by many Au NPs, a collective plasmonic heating effect may exist in the BaTiO₃ and changes the heating process as discussed in Fig. S7f. The authors should clarify this issue.

Reply: If there are multiple optically isolated Au NPs decorated on a single BaTiO₃ particle, the heating effect will be algebraically augmented and the pyro-catalytic H₂ production rate will be further increased. If some of the Au NPs are closely packed to trigger electromagnetic coupling, then the heating effect will be more than simple algebraic summation.

Revision: We add the above discussion in the revised manuscript.

REVIEWERS' COMMENTS

Reviewer #1 (Remarks to the Author):

All relevant aspects were addressed. From my point of view, there is nothing against the publication of the manuscript.

Reviewer #2 (Remarks to the Author):

The authors have responded well to my comments and questions to a large extent. I concede with the author's justification that the yield of their system should not be readily benchmarked to the state of the art of a purely photocatalytic system, as this is a fundamental study and the underlying mechanisms are not the same. If the authors stress this in their manuscript accordingly, this is acceptable.

However, I do retain some concerns with proposing a theoretical upper limit of their system as a key finding, since there is no experimental evidence for this claim whatsoever, and the order of magnitude of this prediction is so vastly different from the first experimental results obtained in this study (three orders of magnitude difference!). Although the estimation has been performed carefully, it remains based on various ideal assumptions (perfect defect-free shapes, perfect homogeneous irradiation of the entire structure, perfect distribution of the plasmonic particles, etc.) that are far from realistic. Preparing and handling such structures in practice will cause an inevitable significant drop in their performance, that is not (and probably cannot) be accounted for at this point in this theoretical estimation. Therefore, even though it is well intended, I still have my reservations concerning the reporting of this 'theoretical maximum performance' as a key finding in the paper, abstract and discussion, as I do not agree it is justified by the results.

Reviewer #3 (Remarks to the Author):

This is an updated version based on my revision request. I can see the authors have fully addressed my concerns by conducting additional experiments and discussions. Thus it can be accepted now.

Response Letter to Reviewers' Comments:

Reviewer #1 (Remarks to the Author):

All relevant aspects were addressed. From my point of view, there is nothing against the publication of the manuscript.

Reply: Thanks.

Reviewer #2 (Remarks to the Author):

The authors have responded well to my comments and questions to a large extent. I concede with the author's justification that the yield of their system should not be readily benchmarked to the state of the art of a purely photocatalytic system, as this is a fundamental study and the underlying mechanisms are not the same. If the authors stress this in their manuscript accordingly, this is acceptable.

Reply: Thanks. We add a sentence “We also want to emphasize that the pyro-catalytic hydrogen production reported here cannot be directly benchmarked to the state-of-the-art photocatalytic hydrogen production from water splitting since the underlying mechanisms are different.” into the revised manuscript.

However, I do retain some concerns with proposing a theoretical upper limit of their system as a key finding, since there is no experimental evidence for this claim whatsoever, and the order of magnitude of this prediction is so vastly different from the first experimental results obtained in this study (three orders of magnitude difference!). Although the estimation has been performed carefully, it remains based on various ideal assumptions (perfect defect-free shapes, perfect homogeneous irradiation of the entire structure, perfect distribution of the plasmonic particles, etc.) that are far from realistic. Preparing and handling such structures in practice will cause an inevitable significant drop in their performance, that is not (and probably cannot) be accounted for at this point in this theoretical estimation. Therefore, even though it is well intended, I still have my reservations concerning the reporting of this 'theoretical maximum performance' as a key finding in the paper, abstract and discussion, as I do not agree it is justified by the results.

Reply: We remove the claim of the theoretical upper limit from the abstract and main text of our manuscript. We just keep the theoretical calculation based on the current geometry of the nanoparticles in the supporting information. The calculated hydrogen generation rate is around three times of the experimental value, which is reasonable.

Reviewer #3 (Remarks to the Author):

This is an updated version based on my revision request. I can see the authors have fully addressed my concerns by conducting additional experiments and discussions. Thus it can be accepted now.

Reply: Thanks.